# Combinatorial deployment of F-actin regulators to build complex 3D actin structures in vivo

**Yi Xie, Rashmi Budhathoki, J Todd Blankenship\***

Department of Biological Sciences, University of Denver, Denver, United States

**Abstract** Despite extensive studies on the actin regulators that direct microfilament dynamics, how these regulators are combinatorially utilized in organismal tissues to generate 3D structures is an unresolved question. Here, we present an in-depth characterization of cortical actin cap dynamics and their regulation in vivo. We identify rapid phases of initiation, expansion, duplication, and disassembly and examine the functions of seven different actin and/or nucleator regulators (ANRPs) in guiding these behaviors. We find ANRPs provide distinct activities in building actin cap morphologies – specifically, while DPod1 is a major regulator of actin intensities, Cortactin is required for continued cortical growth, while Coronin functions in both growth and intensity and is required for Cortactin localization to the cap periphery. Unexpectedly, cortical actin populations recover *more rapidly* after regulator disruption, suggestive of a deep competition for limited G-actin pools, and we measure in vivo Arp2/3 recruitment efficiencies through an ectopic relocalization strategy. Our results illustrate how the coordination of multiple actin regulators can orchestrate organized and dynamic actin structures in a developmental system.

## Introduction

The mechanisms by which complex actin-based structures form are essential to shaping cell and tissue morphologies. This capacity to rapidly direct filamentous actin assembly is key to a cell's ability to either maintain or abruptly distort its cell shape. During development, the rapid changes in tissue morphologies are often a result of the remodeling of cortical actin and myosin activities (reviewed in *Munjal and Lecuit, 2014*; *Heer and Martin, 2017*). In keeping with this importance of regulating actin structures to achieve discrete cell shapes, a multitude of actin regulators are present within the genomes of eukaryotic animals (*Siripala and Welch, 2007*; *Swaney and Li, 2016*; *Pegoraro et al., 2017*). Some of the foremost examples of actin regulators are the nucleation and assembly complexes of the Formin and Arp2/3 complex families. Additionally, there are a host of actin and nucleator regulatory proteins present in the genomes of most higher animals (*Siripala and Welch, 2007*). However, although the biochemical activities of a broad array of actin regulators have been examined in vitro, how these combined activities are utilized by development in vivo to generate three-dimensional structures is less clear. Furthermore, many actin regulators have been implicated in a variety of different processes ranging from the control of filament branching and turnover to the direct regulation and stabilization of nucleator complex function. Thus, the baseline effects of how these proteins contribute to building cortical structures is unclear. Here, we examine the in vivo function of seven major families of actin and/or nucleator regulatory proteins (ANRPs – DPod1, Coronin, Cortactin, Scar, Wasp, Wash, Carmil) in an intact organismal tissue context. We are using the dynamic furrowing processes in the early syncytial fly embryo to study the rapid formation and disassembly of apical actin networks. In the end, the combined activities of different actin regulatory pathways will drive the cell-shaping events necessary for the development and generation of a wide array of cell morphologies.

**\*For correspondence:**
jblanke4@du.edu

**Competing interests:** The authors declare that no competing interests exist.

The *Drosophila* syncytium has a series of rapid transient cleavage cycles that are driven by actin polymerization and membrane trafficking pathways (*Warn, 1986*; *Foe et al., 2000*; *Riggs et al., 2003*; *Pelissier et al., 2003*; *Grosshans et al., 2005*; *Yan et al., 2013*; *Holly et al., 2015*; *Mavor et al., 2016*; *Xie and Blankenship, 2018*; *Zhang et al., 2018*). Following fertilization and the fusion of the male and female pronuclei, the zygotic nucleus undergoes 13 rounds of replication in the absence of cell division to generate a single-celled embryo with approximately 5000 nuclei (*Hartenstein, 1993*; *Mazumdar and Mazumdar, 2002*; *Schmidt and Grosshans, 2018*). The first nine rounds of replication occur deep in the yolk of the embryo; however, at cycle 10, nuclei migrate to the periphery and begin organizing the formation of cortical actin structures at the apical surface (apical F-actin caps) that will then seed the formation of cytokinetic-like furrows that serve to separate mitotic spindles in the syncytium.

Apical actin cap and cleavage furrow behaviors are highly transient, forming during each syncytial cell cycle where they compartmentalize and anchor mitotic spindles, before then regressing (*Foe and Alberts, 1983*; *Sullivan et al., 1993*; *Cao et al., 2010*; *Holly et al., 2015*). Embryos undergo four rapid rounds of actin cap and furrow formation (cycles 10–13), followed by a final fifth round of furrow ingression that results in the permanent packaging of nuclei into individual cells and the formation of an epithelium through a process known as cellularization. The ability to form these apical actin caps and ingressing furrows is essential to genomic stability – when these morphogenetic processes are disrupted, chromosomal segregation defects occur (*Sullivan et al., 1993*; *Holly et al., 2015*; *Xie and Blankenship, 2018*). Cap failure leads to aneuploid and polyploid nuclei, and defective nuclei are subsequently jettisoned into the deep yolk layer where they do not contribute further to development. This illustrates the importance of effective cortical actin cap function in the early embryo. These cleavage cycles of actin and furrow formation are rapid, with each full round of cap assembly and disassembly occurring within 7 (cycle 10) to 20 min (cycle 13). F-actin regulation is therefore highly dynamic, and these stages represent an intriguing system to analyze how three-dimensional forms can be swiftly generated.

Here, we use 4D live-imaging of filamentous actin to determine phasic behaviors during cortical cap formation. Formation of these actin structures progresses through rapid periods encompassing growth, stabilization, elongation, and remodeling activities. Formin and Arp2/3 networks are responsible for building these cortical structures, and individual ANRPs have distinct functions in guiding cap growth and cap-associated actin intensities. We also generate an ANRP-related toolkit of genomic transgenes for these actin regulators and employ a mito-tag strategy to assess the strength of Arp2/3 nucleator recruitment in vivo. Finally, we explore how the disruption of ANRPs leads to *faster* actin network recoveries, which may be suggestive of a competition for a limited G-actin pool that controls actin assembly dynamics in the embryo.

## Results

### Rapid formation and dissolution of apical actin cap structures

The apical actin caps in the early syncytial *Drosophila* embryo are highly active structures that undergo cyclic behaviors of formation and disassembly on relatively short time scales. As such, they represent a unique opportunity to unravel the mechanisms that guide formation of complex three-dimensional actin-based structures. As a starting point, we live-imaged wild-type actin caps by labeling filamentous actin with an actin-binding domain construct derived from moesin (mCh:MoeABD). This method of labeling has been used extensively in the *Drosophila* embryo and well-represents endogenous filamentous actin dynamics while avoiding problems that occur when fluorescent proteins are directly attached to actin or other labeling paradigms (such as Lifeact; *Kiehart et al., 2000*; *Blankenship et al., 2006*; *Spracklen et al., 2014*; *Figure 1—figure supplement 1A–F*). Actin cap formation was imaged through the four cycles that occur in the syncytial blastoderm (nuclear cycles 10–13), and actin growth behaviors were analyzed (*Figure 1A*, *Figure 1—figure supplement 2A–G*; *Figure 1—source data 1*).

Our analysis shows that apical actin caps experience an initial period of rapid exponential growth (~6 fold growth, see Materials and methods). During this phase, the major increase in cap dimensions occurs in as little as 120 s (*Figure 1A–C*, *Figure 1—figure supplement 2*). This is followed by cap stabilization and cap elongation, which correlates with spindle duplication and separation

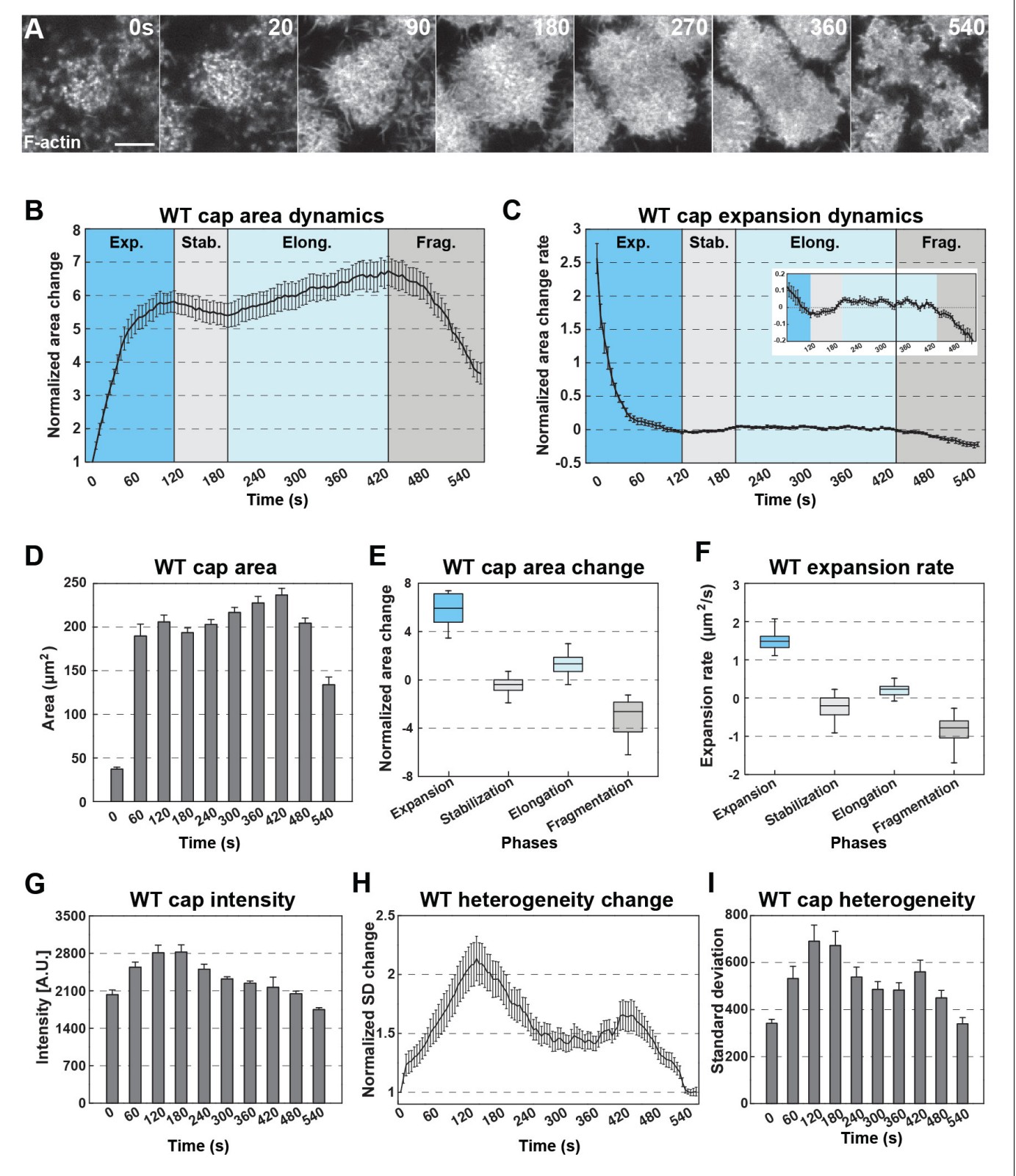

**Figure 1.** Rapid formation and dissolution of cortical actin cap structures. (**A**) Still images from live-imaging of apical F-actin dynamics (UAS:moeABD: mCherry, cycle 11) at t = 0, 20, 90, 180, 270, 360, 540 s. Scale bar = 5 μm. (**B**) WT actin cap area dynamics from cycle 11 (measured cap n = 15, from embryo N = 4). Cap areas are normalized to the size at t = 0 s. Four different phases are labeled (Exp.: *Expansion*; Stab.: *Stabilization*; Elong.: *Elongation*; and Frag.: *Fragmentation* phases). (**C**) WT actin cap expansion rate from 30 s rolling window (cycle 11, n = 15, N = 4). Inset has y-axis re-

*Figure 1 continued on next page*

*Figure 1 continued*

scaled to visualize changes after expansion. (**D**) WT actin cap area ($\mu m^2$) from cycle 11 at t = 0, 60, 120, 180, 240, 300, 360, 420, 480, 540 s (n = 15, N = 4). (**E**) WT actin cap area change in different phases (cycle 11, n = 15, N = 4). The values are calculated by the cap area at the end point divided by the area at the beginning of each phase. (**F**) Average WT actin cap area expansion rate ($\mu m^2$/s) in different phases (cycle 11, n = 15, N = 4). (**G**) Average WT actin cap intensity (A.U.) from cycle 11 at t = 0, 60, 120, 180, 240, 300, 360, 420, 480, 540 s (n = 12, N = 3). (**H**) WT actin cap heterogeneity dynamics from cycle 11 (n = 15, N = 4). The heterogeneity is measured as the intensity standard deviation normalized to the value at t = 0 s. (**I**) WT actin cap heterogeneity from cycle 11 at t = 0, 60, 120, 180, 240, 300, 360, 420, 480, 540 s (n = 15, N = 4).

The online version of this article includes the following source data and figure supplement(s) for figure 1:

**Source data 1.** Cap dynamics data.
**Figure supplement 1.** Similar actin labeling with different actin markers.
**Figure supplement 2.** WT area dynamics of cortical actin caps in cycle 10–13.

(*Figure 1A–C*; *Cao et al., 2010*). Lastly, caps disassemble and then reform as small proto-caps around two central hubs to begin the next cycle of cap behaviors (*Figure 1A,B, Figure 1—figure supplement 1A,B*; *Figure 1—figure supplement 2E–G*). In each cycle, the cap dynamics share similar features, albeit with moderate differences (especially in the very short cycle 10) – as a result, we will focus on cycle 11 caps going forward (*Figure 1—figure supplement 2A–G*).

During the expansion phase, cycle 11 actin caps rapidly expand from small proto-caps to caps 15–18 $\mu m$ in diameter and 206 $\mu m^2$ in area on average within 2 min (*Figure 1A–F*). Apical cap size increases almost sixfold during this period and growth rates peak early during expansion (*Figure 1E, F*). Interestingly, despite the exponential growth in cap size, the actin intensity increases only mildly, with a ~39% increase by the end of stabilization (*Figure 1G*). This suggests that actin recruitment is carefully regulated spatially such that the cap grows rapidly in total area but regional intensities do not.

The following stabilization phase lasts for ~60 s with caps largely maintaining their size. As the cell cycle continues, caps elongate and slightly increase in size (*Figure 1A–F*). However, cap intensities begin to decrease and heterogeneity within the cap drops by approximately one quarter of the maximum at the end of stabilization phase (*Figure 1G–I*). Interestingly, the morphology of the cap during elongation switches from round to an elongated doughnut-like structure and caps begin to lose intensity in internal actin populations (*Figure 1A*). As the cell cycle reaches mitosis and chromosomal segregation, the caps fragment and disassemble. Overall cap area begins to decline as F-actin gradually disbands leaving low-intensity gaps in the middle of the elongating figure and along cap edges (*Figure 1A*, *Figure 1—figure supplement 1A*). During actin cap disassembly, the average intensity of caps drops, as well as the measured heterogeneity (*Figure 1G–I*). These results demonstrate that the apical cap is a dynamic and complex F-actin structure, providing an interesting contextual model for the investigation of filamentous behaviors in vivo.

## Diaphanous and Arp2/3 networks direct actin cap dynamics

After describing the filamentous actin dynamics above, we wanted to examine the major actin networks that drive these behaviors. The Formin Diaphanous and the Arp2/3 complex have been previously implicated in regulating actin nucleation at these syncytial stages (*Stevenson et al., 2002*; *Zallen et al., 2002*; *Grosshans et al., 2005*; *Cao et al., 2010*; *Zhang et al., 2018*). According to published studies, Diaphanous appears to be the major regulator of furrow-associated F-actin, while Arp2/3 has been implicated in apical actin formation (*Grosshans et al., 2005*; *Cao et al., 2010*; *Zhang et al., 2018*), although comprehensive time-lapse-based quantitation has been lacking. We therefore performed our own quantitative analysis of these protein's function in the early syncytial stages. Consistent with previous results, disruption of Diaphanous function deeply affects furrow-associated actin (*Figure 2—figure supplement 1*). However, Diaphanous also shows a significant contribution to the early expansion of actin caps. *Dia* disrupted embryos have a ~ 35% reduction in cap area expansion, but a relatively mild 18% reduction in actin intensities and cap expansion rate (*Figure 2A–H*, *Figure 2—source data 1*). We then examined the contribution of the Arp2/3 complex to actin behaviors. By contrast to Diaphanous, when Arp2/3 function is compromised there is an almost complete absence of cap expansion and cap actin intensities are reduced to 47% of wild-type levels (*Figure 2A–H*). Interestingly, the remaining actin structure appear to be hollowed out and missing internal actin populations (*Figure 2A,B*). As recent evidence has shown that Arp2/3 and

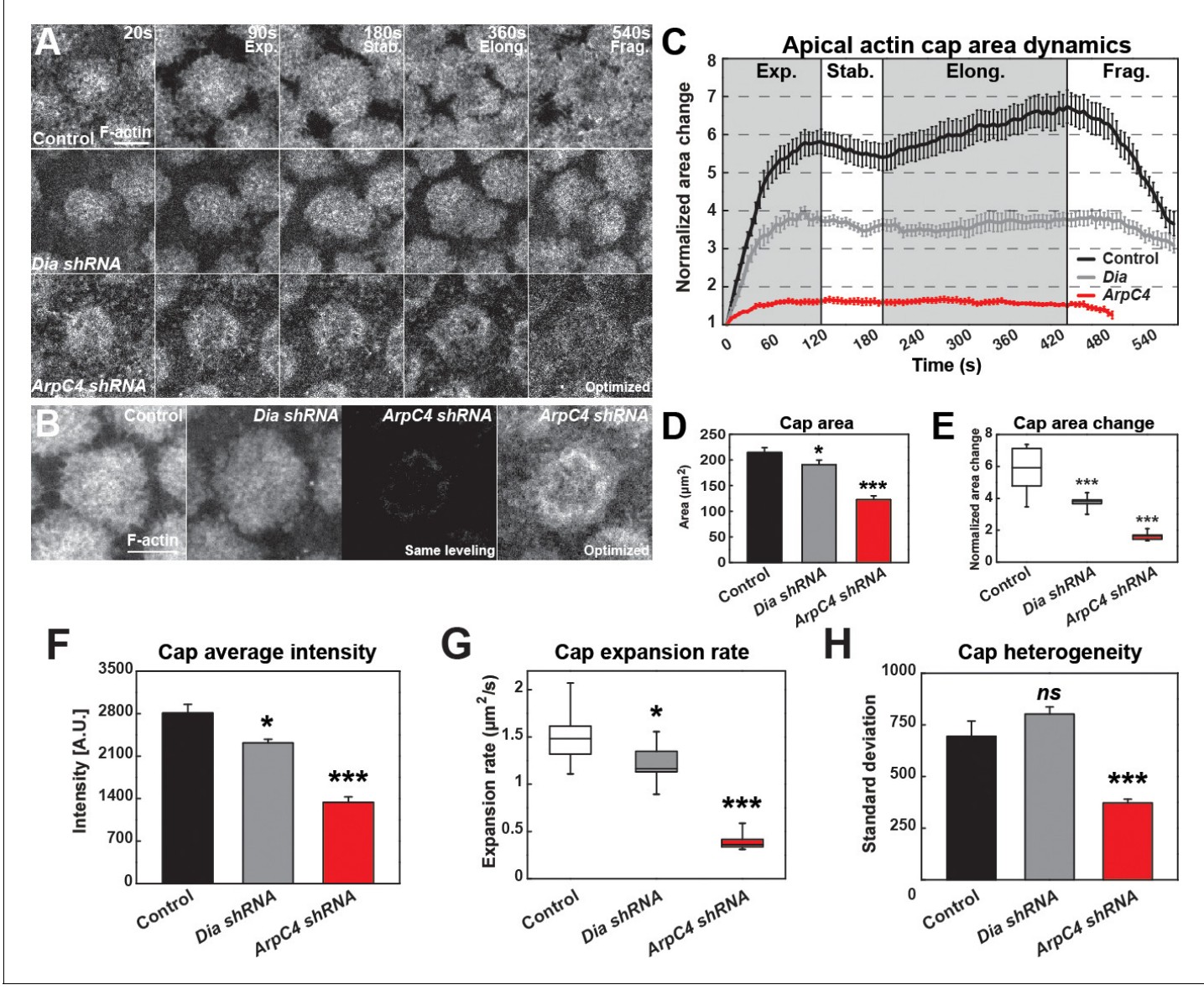

**Figure 2.** Quantitative dynamics of Formin and Arp2/3-driven actin networks. (**A**) Still images from live-imaging of apical F-actin dynamics (UAS: moeABD:mCherry, cycle 11) from control, *Dia shRNA* and *ArpC4 shRNA* lines at t = 20, 90, 180, 360, 540 s. Four different phases are labeled (Exp.: *Expansion*; Stab.: *Stabilization*; Elong.: *Elongation*; and Frag.: *Fragmentation* phases). Scale bar = 5 μm. (**B**) Still images showing F-actin cap intensities by live-imaging (UAS:moeABD:mCherry, cycle 11) from control, *Dia shRNA* and *ArpC4 shRNA* lines at t = 120 s. First three panels are leveled and imaged equivalently, with the last panel optimized for visualization. Scale bar = 5 μm. (**C**) Actin cap area dynamics of control (black, n = 15, N = 4), *Dia shRNA* (grey, n = 10, N = 3) and *ArpC4 shRNA* (red, n = 11, N = 3) from cycle 11. Cap areas are normalized to the size at t = 0 s. (**D**) Actin cap area (μm$^2$) of control (n = 15, N = 4), *Dia shRNA* (n = 10, N = 3), and *ArpC4 shRNA* (n = 11, N = 3) at t = 120 s in cycle 11. *: $p<0.05$, ***: $p<0.0005$. (**E**) Actin cap area change of control (n = 15, N = 4), *Dia shRNA* (n = 10, N = 3), and *ArpC4 shRNA* (n = 11, N = 3) from t = 120 s to t = 0 s in cycle 11. ***: $p<0.0005$. (**F**) Average intensity of apical cap structures of control (n = 12, N = 3), *Dia shRNA* (n = 10, N = 3), and *ArpC4 shRNA* (n = 11, N = 3) at t = 120 s in cycle 11. *: $p<0.05$, ***: $p<0.0005$. (**G**) Actin cap area expansion rate of control (n = 15, N = 4), *Dia shRNA* (n = 10, N = 3), and *ArpC4 shRNA* (n = 11, N = 3) from 0 to 120 s in cycle 11. *: $p<0.05$, ***: $p<0.0005$. (**H**) Actin cap heterogeneity (intensity standard deviation) of control (n = 15, N = 4), *Dia shRNA* (n = 10, N = 3), and *ArpC4 shRNA* (n = 11, N = 3) at t = 120 s in cycle 11. *ns*: not significant, ***: $p<0.0005$.

The online version of this article includes the following source data and figure supplement(s) for figure 2:

**Source data 1.** Arp and Dia disrupted data.

**Figure supplement 1.** Apical and furrow-associated actin populations in *Dia* or *Arp2/3* compromised embryos.

Formin networks can demonstrate a degree of interdependence (*Suarez et al., 2015*; *Chan et al., 2019*), we examined Diaphanous and Arp2/3 intensities when each network was disrupted. Although defective cap structures characteristic of the above described results were observed, disruption of *Diaphanous* function did not significantly affect Arp2/3 intensities (as judged by ArpC1:GFP), nor did *Arp2/3* disruption affect anti-Diaphanous intensities in the remaining cap structures (anti-Diaphanous immunostaining; *Figure 2—figure supplement 1B–E*). These results show that Diaphanous and Arp2/3 both contribute to structuring apical actin caps, but that Arp2/3 is the major regulator of actin intensities and cap growth. The hollowed out internal cap structures as well as the failure in cap expansion also suggests a possible model in which Arp2/3 polymerizes F-actin internally, and the cap possesses a dispersion mechanism that flows F-actin filaments towards the edges so as to maintain consistent internal F-actin intensities.

## Different ANRPs have distinct roles in building apical actin structures

Given the above rapid dynamics of cap formation and nucleator function, we wanted to identify how actin and/or nucleator regulatory proteins (ANRPs) are deployed to control actin activities spatio-temporally in the construction of cortical actin caps. Extensive work across a variety of systems has revealed a diverse array of actin regulatory proteins, many of which have also been implicated, to varying degrees, in controlling Arp2/3 activity or stability, although additional actin-related functions for these proteins exist (*Siripala and Welch, 2007*; *Swaney and Li, 2016*). Here we are focusing on seven of these families of proteins (DPod1, Coronin, Cortactin, Scar, Carmil, Wasp, and Wash) – each of these families is represented within the *Drosophila* genome by single orthologs (*Figure 3A*). Cortactin and Carmil have been shown to regulate Arp2/3 complex function as well as branch point stability (Cortactin) or filament capping (Carmil), and Scar/Wash/Wasp super-family proteins are known potent activators of Arp2/3 nucleation (*Jung et al., 2001*; *Uruno et al., 2001*; *Weaver et al., 2001*; *Ammer and Weed, 2008*; *Pollitt and Insall, 2009*). Scar has previously been suggested to be the most relevant member regulating Arp2/3 function in the early embryo, although this has not been tested systematically (*Zallen et al., 2002*; *Levayer et al., 2011*). *Drosophila* DPod1 contains WD40 domains and has similarity to Coronin-family proteins, which have been implicated in recruiting Arp2/3 complexes in the presence of preexisting actin filaments, as well as regulating cofilin function (*Gandhi and Goode, 2008*). Given the many associated functions of the ANRPs, we wanted to determine their baseline behaviors in regulating in vivo actin structures, and therefore analyzed actin cap dynamics in embryos compromised for each of these ANRPs (*Figure 3B*, *Figure 3—source data 1*).

Interestingly, these results identify distinct functions for Cortactin, DPod1, Coronin, and Scar in building F-actin caps (*Figure 3D–K*), while disrupting Carmil, Wasp, and Wash had little effect on caps (*Figure 3L,M, Figure 3—figure supplement 1J–M*). Compromising Coronin function causes an immediate defect in the expansion phase of cap formation (*Figure 3G,G'*), with actin intensities at ~70% of control levels (*Figure 3C,F,G''*). By contrast, disrupting DPod1 function produces caps that expand at near wild-type levels and possess wild-type areas until they fail to maintain area size in the later stages of apical cap function (*Figure 3E,E'*). However, these embryos have dramatically reduced actin intensities throughout the cap area (*Figure 3E''*). In embryos with compromised Cortactin function, there is a normal burst of actin expansion, but after ~120 s actin caps do not continue to grow and steadily diminish in size, suggesting a role for Cortactin in growth at the cap periphery (*Figure 3H,I–I''*). Interestingly, F-actin intensities within the smaller cap are at *higher* levels than control embryos (*Figure 3I''*). Disrupting Scar function produces actin caps that show an early depletion of F-actin intensities followed by a delayed expansion phase in which the caps cannot fully reach control cap areas (*Figure 3J,K–K''*). The specificity and reproducibility of these phenotype was confirmed with second, independent shRNA lines (*Figure 3—figure supplement 1*).

Given these effects on the size, shape, and intensity of apical actin caps, and our results demonstrating that the Arp2/3 complex is the major regulator of actin behaviors in the apical cortex, we next determined the degree to which Arp2/3 complex recruitment (as proxied by an endogenous CRISPR Arp3:GFP) to the apical cortex was compromised in these various backgrounds. Disrupting the Formin Diaphanous had no effect on Arp3:GFP localization at apical structures (*Figure 3N,O*). By contrast, disruption of ArpC4, one of the Arp2/3 complex subunits, almost completely abolishes apical Arp3:GFP localization and intensity (*Figure 3N,O*). Interestingly, DPod1, which had the deepest impact on overall F-actin intensities, also had the largest effect on Arp3:GFP localization, while

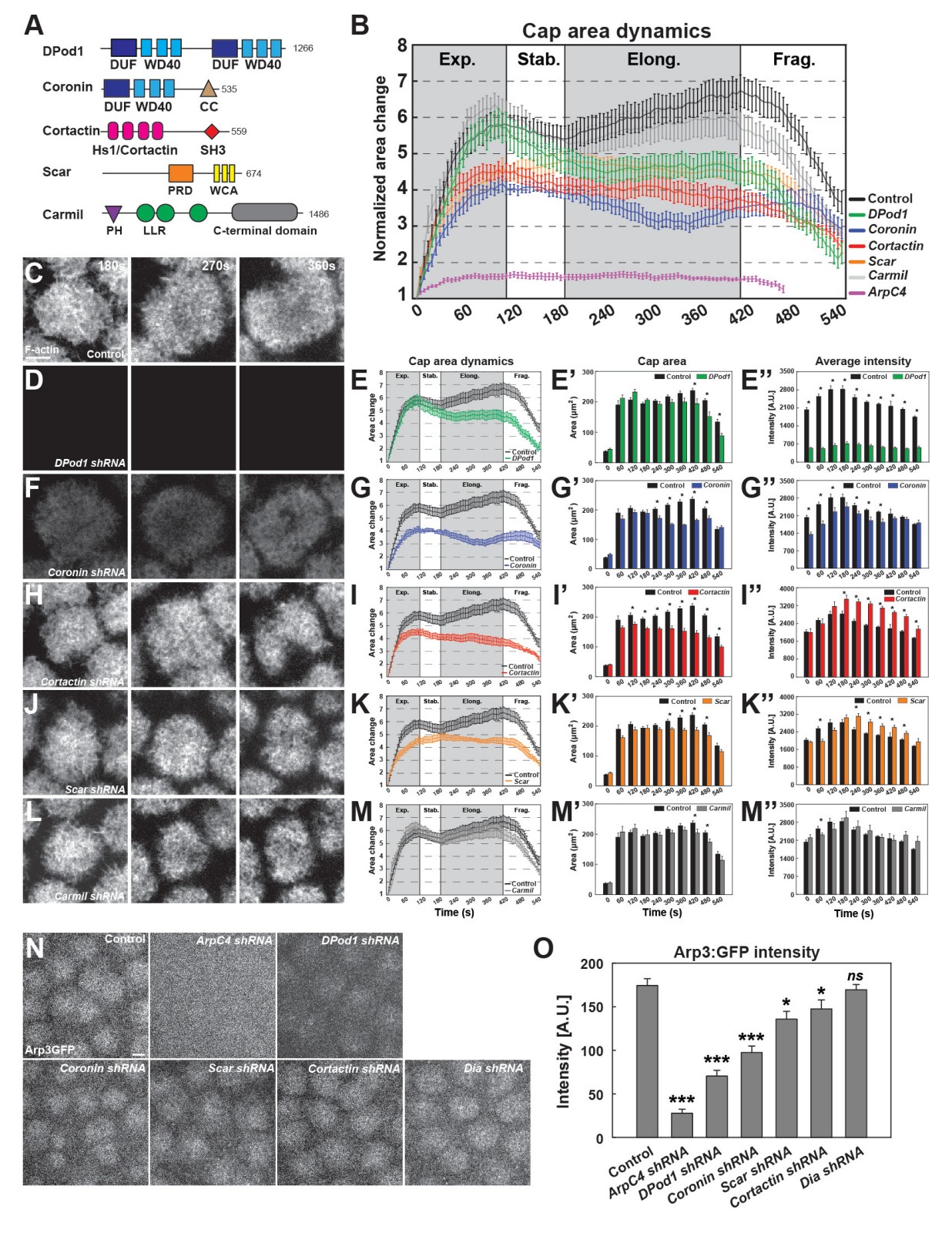

**Figure 3.** ANRPs have distinct roles in building cortical actin structures. (**A**) Schematics of different ANRPs domain organization. Domains are collected from Flybase (Pfam/SMART) and UniProt. WD40: WD40 repeats; CC: coiled-coil domain; Hs1/Cortactin: Hs1/Cortactin repeats; SH3: SH3 domain; PRD: proline-rich domain; WCA: WH2/verprolin, cofilin, acidic domains; LRR: Leucine-rich repeats; PH: pleckstrin homology domain; LLR: Leucine-rich repeats; C-terminal: Carmil c-terminal domain; DUF: domains of unknown function. Domain size is not to scale. (**B**) Cycle 11 apical actin cap area

*Figure 3 continued on next page*

Figure 3 continued

dynamics in control, Arp2/3, and ANRP disrupted embryos: control (black, n = 15, N = 4), *DPod1 shRNA* (green, n = 11, N = 3), *Coronin shRNA* (blue, n = 9, N = 3), *Cortactin shRNA* (red, n = 12, N = 3), *Scar shRNA* (orange, n = 11, N = 3), *Carmil shRNA* (grey, n = 9, N = 3), and ArpC4 shRNA (magenta, n = 11, N = 3). Cap areas are normalized to the size at t = 0 s. (C, D, F, H, J, L) Still images from live-imaging of apical F-actin dynamics (UAS:moeABD:mCh, cycle 11) at t = 180, 270, 360 s, from control (C), *DPod1 shRNA* (D), *Coronin shRNA* (F), *Cortactin shRNA* (H), *Scar shRNA* (J), and *Carmil shRNA* (L) embryos. Images are identically leveled and imaged. Scale bar = 5 μm. (E, G, I, K, M) Apical actin cap area dynamics (cycle 11) in control (black, n = 15, N = 4), *DPod1 shRNA* (green, n = 11, N = 3), *Coronin shRNA* (blue, n = 9, N = 3), *Cortactin shRNA* (red, n = 12, N = 3), *Scar shRNA* (orange, n = 11, N = 3), *Carmil shRNA* (gray, n = 9, N = 3), and ArpC4 shRNA (magenta, n = 11, N = 3). Cap areas are normalized to the size at t = 0 s. (E', G', I', K', M') Absolute actin cap areas (μm$^2$) in control and ANRP-compromised embryos from cycle 11 at indicated time points. *: p<0.05. (E'', G'', I'', K'', M'') Actin cap average intensity in control and ANRP-compromised embryos from cycle 11 at indicated time points. *: p<0.05. Bar graphs without * labeled in (E'–E'', G'–G'', I'–I'', K'–K'', M'–M'') are not significant. (N) Still images of endogenous CRISPR Arp3:GFP behavior in control and actin regulator disrupted embryos at t = 120 s in cycle 11. Scale bar = 5 μm. (O) CRISPR Arp3:GFP intensity in control and different actin regulators functional disruption embryos at t = 120 s in cycle 11. control: n = 13, N = 3; ArpC4 shRNA: n = 18, N = 3; *DPod1 shRNA*: n = 34, N = 3; *Coronin shRNA*: n = 30, N = 3; *Scar shRNA*: n = 24, N = 3; *Cortactin shRNA*: n = 27, N = 3; *Dia shRNA*: n = 28, N = 3. ns: not significant, *: p<0.05, **: p<0.005; ***: p<0.0005.

The online version of this article includes the following source data and figure supplement(s) for figure 3:

**Source data 1.** ANRP measurement data.
**Figure supplement 1.** ANRPs have distinct roles in building apical actin structures.

Coronin, Scar, and Cortactin showed intermediate Arp3:GFP recruitment defects. Together, these results reveal distinct functions for individual ANRPs, with DPod1 being required for overall actin intensities, while Cortactin and Scar are necessary for cap expansion and maintenance. These results are also consistent with Coronin having an early function, possibly in the cap center, and Cortactin/Scar possessing a later function in controlling cap growth and maintenance of cap areas at the periphery. Coronin and Scar have both intensity (at early phases) and cap size functions, and Carmil, Wasp, and Wash have either minor or no contributions to cap dynamics.

## Coronin directs Cortactin localization to the cap periphery

To examine the expression and localization of the ANRPs during cap formation, we first used qPCR to examine the relative mRNA levels of each ANRP in the early syncytium. These results show that Coronin, Cortactin, and DPod1 are highly expressed and Carmil and Wash are present at low levels during syncytial stages (*Figure 4—figure supplement 1G*). Dia, Wasp, and Scar have intermediate expression. To further investigate the localization of each regulator, we generated an ANRP toolkit of expression constructs. We first generated either N- or C-terminal UAS GFP expression transgenic constructs (and, in many cases, both N- and C- terminal; *Table 1*), and then followed up with either CRISPR/Cas9-mediated homologous recombination to knock-in GFP at endogenous loci or generated antibodies to examine endogenous localization. A CRISPR-generated GFP knock-in at the endogenous Arp3 locus shows a strong localization to the apical cap with little localization to furrows, consistent with our functional data (*Figure 3N*, *Figure 4—figure supplement 1A*). Neither UAS-driven Carmil nor Wasp localized to apical actin caps (*Figure 4E*, *Figure 4—figure supplement 1E*), again consistent with the above functional analysis indicating their disruption had little impact on apical cap dynamics. UAS-driven Coronin, Cortactin, Scar, and DPod1 all displayed varying degrees of localization to actin caps (*Figure 4A–D*, *Figure 4—source data 1*), with endogenous CRISPR constructs or antibody stains showing similar patterns (*Figure 4—figure supplement 1B–D*). One intriguing facet of these regulators' localization, however, is that Coronin and Cortactin possessed a complementary localization in mature caps, with Cortactin enrichment occurring at the cap periphery, while Coronin possesses an enrichment in the cap interior (temporal overlays in *Figure 4F–I*). This is also consistent with our functional analysis, in which Cortactin was required for cap growth late in the expansion phase and during later size maintenance, while Coronin was required for early cap growth. The complementary localization also suggests a possible antagonism between Coronin and Cortactin. To examine this, we imaged GFP:Cortactin embryos when Coronin was disrupted. Remarkably, this revealed that, in the absence of Coronin function, Cortactin fails to transition to the cap periphery, indicating that Coronin contributes to the ability of Cortactin to localize to the cell periphery to direct actin growth and the maintenance of cap edges (*Figure 4J,K*).

**Table 1.** Arp2/3 and ANRPs toolkit.

| Construct | Vector | Chromosome |
|---|---|---|
| CRISPR Arp3:GFP | Endogeous | III |
| CRISPR GFP:Cortactin | Endogeous | III |
| CRISPR GFP:DPod1 | Endogeous | X |
| UAS:GFP:Cortactin | pUAST | X, II, III |
| UAS:Cortactin:GFP | pUAST,pUASp | X, II, III |
| UAS:mCherry:Cortactin | pUASp | II, III |
| UAS:DPod1:GFP | pUAST | X, II, III |
| UAS:Coronin:GFP | pUAST | II, III |
| UAS:GFP:Carmil FL | pUAST | X, II, III |
| UAS:Carmil FL:GFP | pUAST | X, II, III |
| UAS:Scar:GFP | pUASp | II, III |
| UAS:mito:mCherry:Cortactin | pUASp | II, III |
| UAS:mito:mCherry:Coronin | pUASp | X, II, III |
| UAS:mito:mCherry:DPod1 | pUASp | II, III |
| UAS:mito:mCherry:Scar | pUASp | II, III |
| UAS:mito:mCherry | pUASp | II, III |

## *Faster* recovery of F-actin networks after ANRP disruption

We next examined the dynamics of how actin networks in the early embryo form. To do so, we analyzed recovery rates after photobleaching. As a starting point, we measured recovery in cortical cap populations and in furrow-associated actin populations at ~120 s (when caps are approaching their early maximum in size) into cycle 11. These results revealed that actin is highly dynamic, with a half-time of recovery (T50) of only 8.2 s in the cap and a low immobile fraction of 17% (*Figure 5A–D*, *Figure 5—source data 1*). Furrow-associated actin is more stable with a T50 of 15.1 s (*Figure 5C*). These recovery rates are nearly identical when actin is directly labeled with GFP, again demonstrating that tracking actin cap behaviors with the MoeABD:GFP accurately reflects actin dynamics and that any on–off actin-binding rates by MoeABD:GFP are significantly slower than actin turnover rates (*Figure 1—figure supplement 1D–F*).

We then measured recovery rates when the two major actin nucleating factors in the early embryo, Arp2/3 and Diaphanous, are disrupted. Our expectation was that as these networks are essential for cap growth and actin intensities, we would observe a longer recovery time after photobleaching. Surprisingly, we found that actin recovery is *much faster* when either Arp2/3 or Diaphanous function is disrupted. Indeed, although actin intensities are much reduced, the half-time to recovery of these intensities is nearly twice as fast as in wild-type embryo (4.2 s and 4.1 s in *ArpC4* and *Diaphanous shRNA* embryos, respectively; *Figure 5E*). To examine this further, we analyzed what would happen to recovery rates when more actin is bound into stable filaments and less G-actin is available. We therefore injected embryos with low levels of jasplakinolide to stabilize F-actin – under these conditions, recovery times increased (*Figure 5—figure supplement 1C–D*). Similarly, reducing the G-actin availability by low-dose Latrunculin B injection also slowed recovery times (*Figure 5—figure supplement 1C–D*). These results are intriguing, and at least two potential models could explain these behaviors: (1) the enhanced FRAP recovery times are driven by increased rates of nucleation and polymerization due to higher free G-actin pools or (2) higher FRAP recovery rates could be caused by increased actin turnover. Given that both Diaphanous and Arp2/3 complex function are deeply implicated in directing actin growth and nucleation, and the fact that they are chronically depleted in these experiments, we would suggest that these results support the first model and argues that these complexes are in a strong competition for a limited pool of available G-actin monomers. Consistent with this, FRAP analysis after disrupting Cofilin function (known as *twinstar* in *Drosophila*) did not significantly slow down FRAP recovery times (*Figure 5—figure supplement 1A,B*).

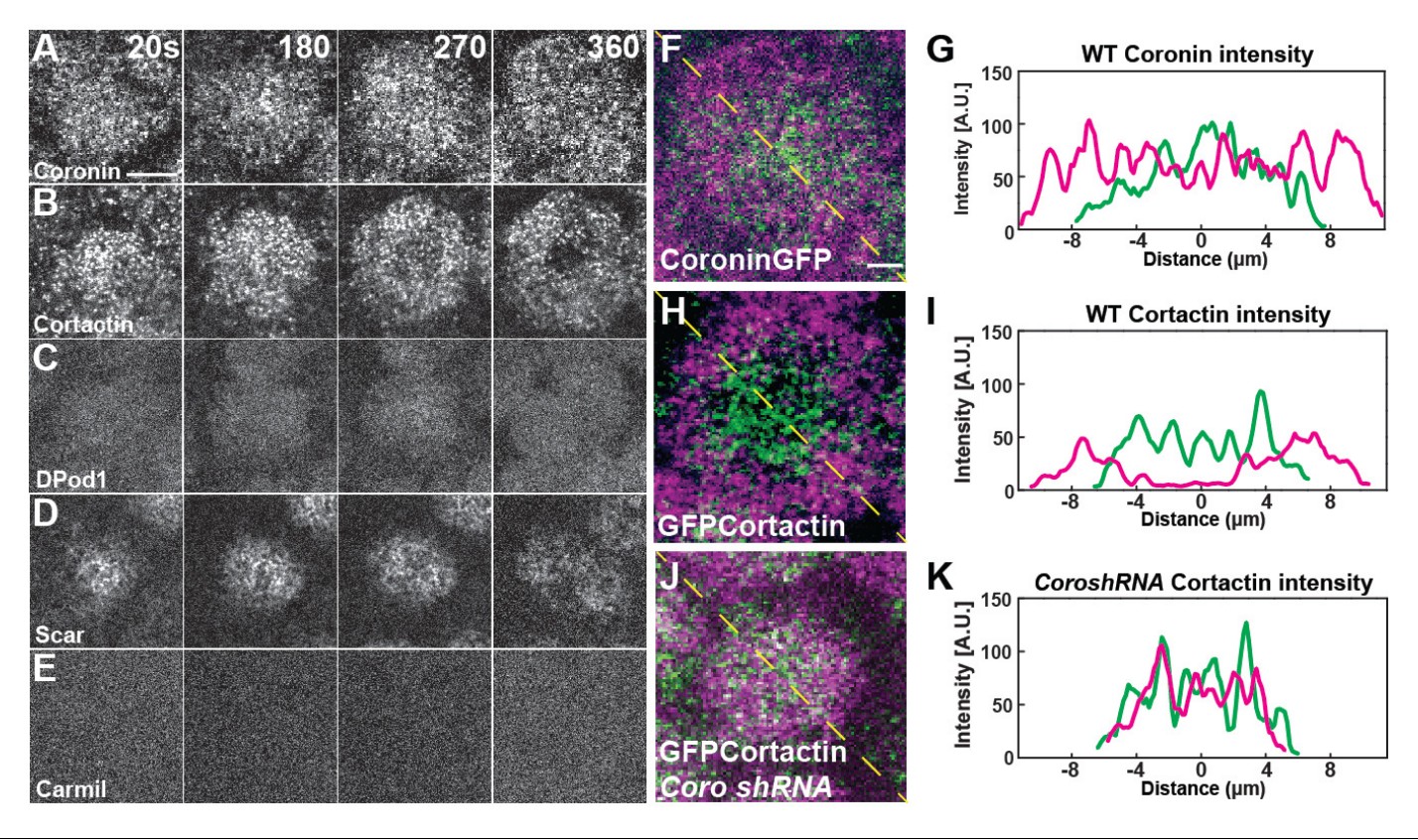

**Figure 4.** ANRP localization at apical actin caps. (**A–E**) Coronin (UAS:Coronin:GFP), Cortactin (UAS:GFP:Cortactin), DPod1 (UAS:DPod1:GFP), Scar (UAS: Scar:GFP), and Carmil (UAS:Carmil:GFP) localization on apical cap structures at t = 20, 180, 270, and 360 s. *Additional Cortactin CRISPR and DPod1 CRISPR allele and anti-Coronin immunostaining data in Fig. Supplement 4-1.* Scale bar = 5 μm. (**F–H**) Cortactin transitions to actin periphery through Coronin antagonism during cap growth. Overlapped images from t = 360 s (magenta) to t = 180 s (green) from live-imaging. Coronin (**F**) and Cortactin (**H**) images are derived from live-imaging of UAS:Coronin:GFP (**F**) and CRISPR GFP:Cortactin (Fig S6B), respectively. Scale bar = 2 μm. (**J**) CRISPR GFP: Cortactin t = 180 s (green) and 360 s (magenta) overlapped images in *Coronin shRNA* disrupted embryo. Scale bar = 2 μm. (**G, I, K**) Intensity profiles from (**F**), (**H**), and (**J**) yellow lines, respectively. Distance = 0 μm indicates the center of apical cap structures.

The online version of this article includes the following source data and figure supplement(s) for figure 4:

**Source data 1.** RT PCR source data.

**Figure supplement 1.** Expression levels and endogenous localization data of ANRPs.

---

Finally, we measured cap recovery rates when the individual ANRPs were disrupted. Similar to disrupting Arp2/3 or Dia function, compromising DPod1, Coronin, Cortactin, or Scar led to faster recovery times, although to varying degrees (*Figure 5—figure supplement 1A–B,E*). Interestingly, DPod1 disruption was almost comparable to disrupting Arp2/3 function in its effect on recovery times and immobile fractions. This is consistent with our data indicating that DPod1 has the strongest impact on F-actin intensities in the cap. By analogy to the above, this could also suggest that actin intensities in the various disrupted backgrounds appear to well-reflect the degree to which specific actin networks are the predominant G-actin utilizing networks in the embryo.

## Nucleator recruitment strengths of ANRP regulators

As we have examined the localization and function of the different ANRPs in cortical cap formation, and as the Arp2/3 complex is the major regulator of new actin in the apical cortex, we wanted to test the strength of Arp2/3 recruitment by each ANRP in vivo. Although, as discussed above, the ANRPs have been implicated in several different mechanisms of actin-regulation, many of the ANRPs have been shown to either activate or stabilize Arp2/3 complex function (*Uruno et al., 2001*; *Weaver et al., 2002*; *Uetrecht and Bear, 2006*; *Pollitt and Insall, 2009*; *Bhattacharya et al.,*

*2016*). We therefore adapted a mitochondrial-tagging assay (*Shin et al., 2020*) to recruit ANRPs to the mitochondria and then tested the degree to which Arp3 and F-actin become ectopically localized. Since DPod1, Cortactin, Coronin, and Scar had the strongest effects on actin cap formation, we fused each of these ANRPs to mCherry and an outer mitochondrial membrane mito-tag (Tom70-HA, 58 amino acids). Intriguingly, the mito-tagged ANRPs are each capable of recruiting Arp3:GFP, although an ANRP-less mito-tag control did not (*Figure 6A–J*, *Figure 6—source data 1*). Furthermore, they also appear able to activate Arp2/3 complex function, as filamentous actin is observed at the mito-tag puncta (*Figure 6—figure supplement 1B–E*). To measure the strength of recruitment, we quantified the colocalization percentage and relative Arp3:GFP intensity as normalized to mito-ANRP:mCherry intensity. Of the tested ANRPs, mito-DPod1 possessed the strongest colocalization and recruitment ability (*Figure 6C–D*). By contrast, Coronin had the lowest colocalization and recruitment ability, while Cortactin and Scar had intermediate Arp3-recruiting activities (*Figure 6E–J*). These results are consistent with our functional analysis, which indicated that DPod1 is most important for overall actin cap intensities and suggests that a high potency ANRP, DPod1, has been selected to drive overall actin levels, while Coronin, Cortactin, and Scar largely have a spatial function in driving cap expansion. These results also provide some insight on the ANRPs relative potencies, as Cortactin demonstrates a high percent of colocalization, but relatively low recruitment ability, while Scar has moderate Arp3 colocalization but high recruitment ability (*Figure 6O, P*). Although it was not possible to recover flies that had pair-wise mito-tag combinations of every mito-ANRP, we were able to analyze embryos that possessed both Coronin and Cortactin mito-tags, as well as embryos that had both DPod1 and Scar mito-tags. Intriguingly, the Arp3 localization and recruitment ability was very low in the double mito-Coro+mito-Cort, while Arp3 recruitment was very high in the double mito-DPod1+mito-Scar background (*Figure 6O,P*). This is again consistent with a potential inhibitory interaction between Coronin and Cortactin and with DPod1 and Scar being potent activators of Arp2/3 recruitment and activity.

## ANRPs function in building cap properties essential for apical nuclear anchoring

Finally, we wanted to determine what the physiological impact of having cortical actin caps with different expansion rates, sizes, and intensities would be on development. As F-actin caps have been implicated in the apical anchorage and positioning of nuclei (*Foe and Alberts, 1983*; *Sullivan et al., 1993*; *Blankenship and Wieschaus, 2001*), we examined what the critical actin properties are that mediate nuclear anchorage against the substantial mitotic flows during division cycles. Loss of apical anchorage is readily apparent in Arp2/3 compromised embryos, with multiple nuclear fallout events being observed in a single cycle (*Figure 7A*, *Figure 7—source data 1*). In previous work, we have shown that a failure to properly segregate chromosomes led to aneuploid or polyploid nuclei and subsequent loss of apical nuclear positioning (*Xie and Blankenship, 2018*). However, here a different mechanism is at work, as we observed that nuclei that underwent apparently normal cell divisions still lost apical positioning in the Arp2/3 compromised background (*Figure 7A*). We therefore correlated nuclear fallout with cortical cap properties such as intensity and cap expansion rates in the various ANRP backgrounds. This analysis revealed that the key property for nuclear anchorage was the growth in cap areas (*Figure 7B*), while overall cap intensities had little correlation to nuclear fallout rates (*Figure 7C*). These data suggest that cortical actin cap expansion and organization, as mediated by Cortactin, Coronin, and Scar, are essential for nuclear positioning and the maintenance of apical nuclear-cortex attachment sites.

## Discussion

Cells have a variety of actin regulatory proteins to select from in the construction of cortical structures that support cell shape and function. Here, we used the early *Drosophila* syncytium as a system to study rapidly developing actin structures and tested the function of seven different ANRP family members in directing specific properties of the apical cortex. We created an ANRP toolkit composed of 18 different transgenic constructs (*Table 1*) to analyze the interplay of actin regulators in an intact morphogenetic organism. This toolkit should form a useful reagent collection for the fly community and has revealed that unique ANRPs were used to drive specific aspects of the growing actin cap. We observed that DPod1 has an essential function in supporting the overall actin intensities in the

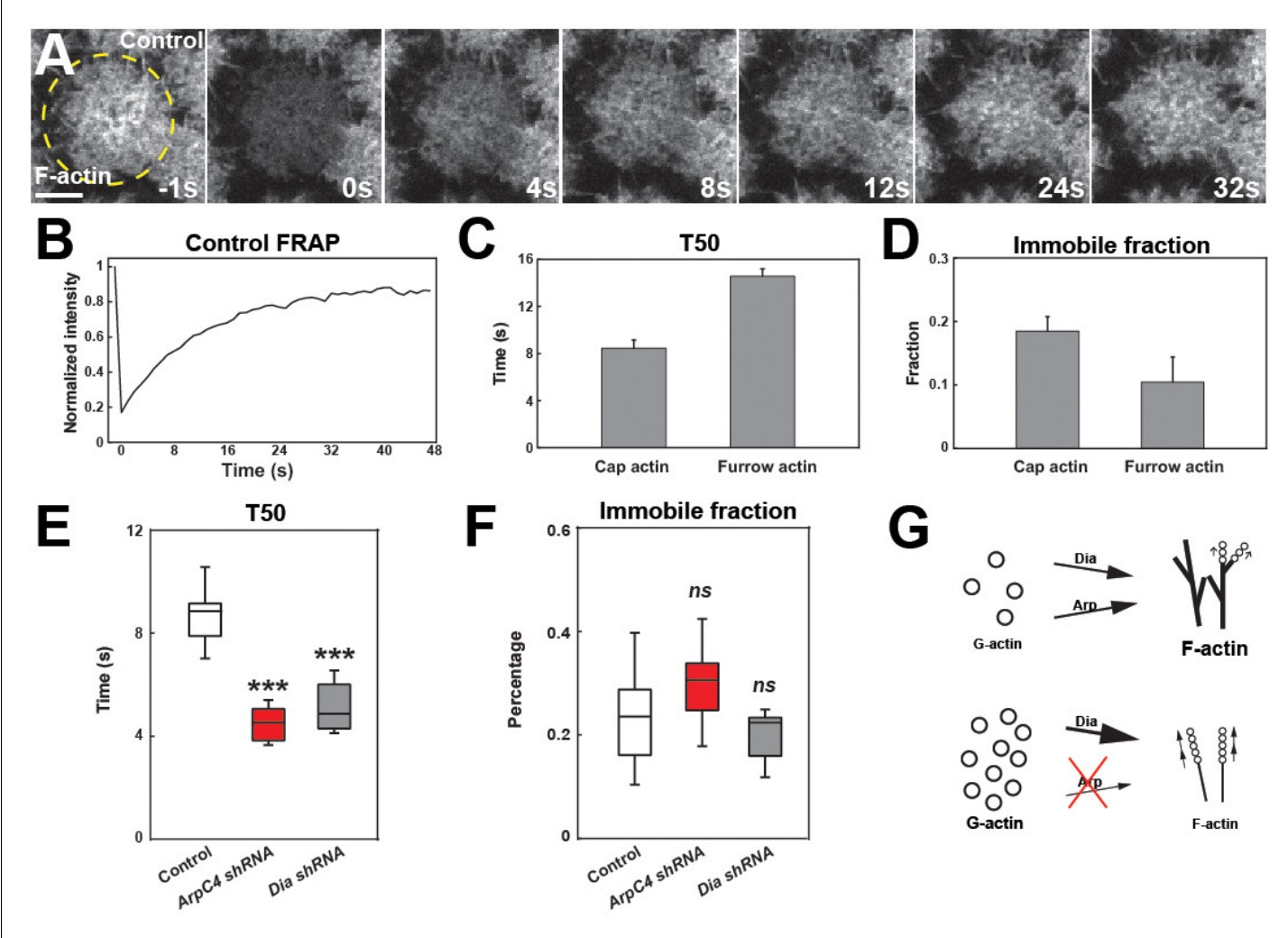

**Figure 5.** Arp2/3 and Formin network analysis suggests competition for free G-actin. (**A**) Still images from FRAP recovery of F-actin live-imaging (UAS: moeABD:mCh) in cycle 11 embryos at t = −1s (1 s before photobleaching), 0 s (photobleaching), 4 s, 8 s, 12 s, 24 s, and 32 s. Scale bar = 5 µm. (**B**) FRAP recovery dynamics in control apical actin cap at cycle 11. Intensity is normalized to the value at t = −1s. (**C, D**) FRAP T50 and immobile fraction of F-actin at apical cap (n = 6) and furrow (n = 3) structures in cycle 11. (**E, F**) FRAP T50 and immobile fraction of F-actin at apical cap structures from control (n = 13), *ArpC4 shRNA* (n = 14), and *Dia shRNA* (n = 9) in cycle 11 embryos showing faster actin recovery rates in *ArpC4* and *Dia* shRNA embryos. ns: not significant, ***: p<0.005. (**G**) Schematic of Arp2/3 and Dia competition for limited G-actin pool.

The online version of this article includes the following source data and figure supplement(s) for figure 5:

**Source data 1.** FRAP actin network source data.

**Figure supplement 1.** FRAP on Arp2/3 and ANRPs disrupted embryos.

cap, but does not appear to function in directing the expansion of the actin cap. In contrast, Cortactin does not contribute to actin intensities, but plays a key role driving the continued growth and expansion of the cap. Interestingly, Coronin, which shares similar WD40 and DUF domain architectures to DPod1, has a dual role in supporting both actin intensity and cap growth. Coronin also shows the earliest function in directing cap growth, while Cortactin and Scar have cap growth rates that become compromised during the late portions of the expansion phase. Interestingly, our results also showed a potential antagonism between Cortactin and Coronin that may underlie these early and late functions of the two regulatory proteins. Cortactin localizes to the cap periphery in later cortical caps, but fails to undergo this transition when Coronin function is disrupted. This suggests that centrally located Coronin may aid in directing Cortactin to a peripheral enrichment, and is consistent with a previous study showing a competition between Coronin and Cortactin in binding at actin

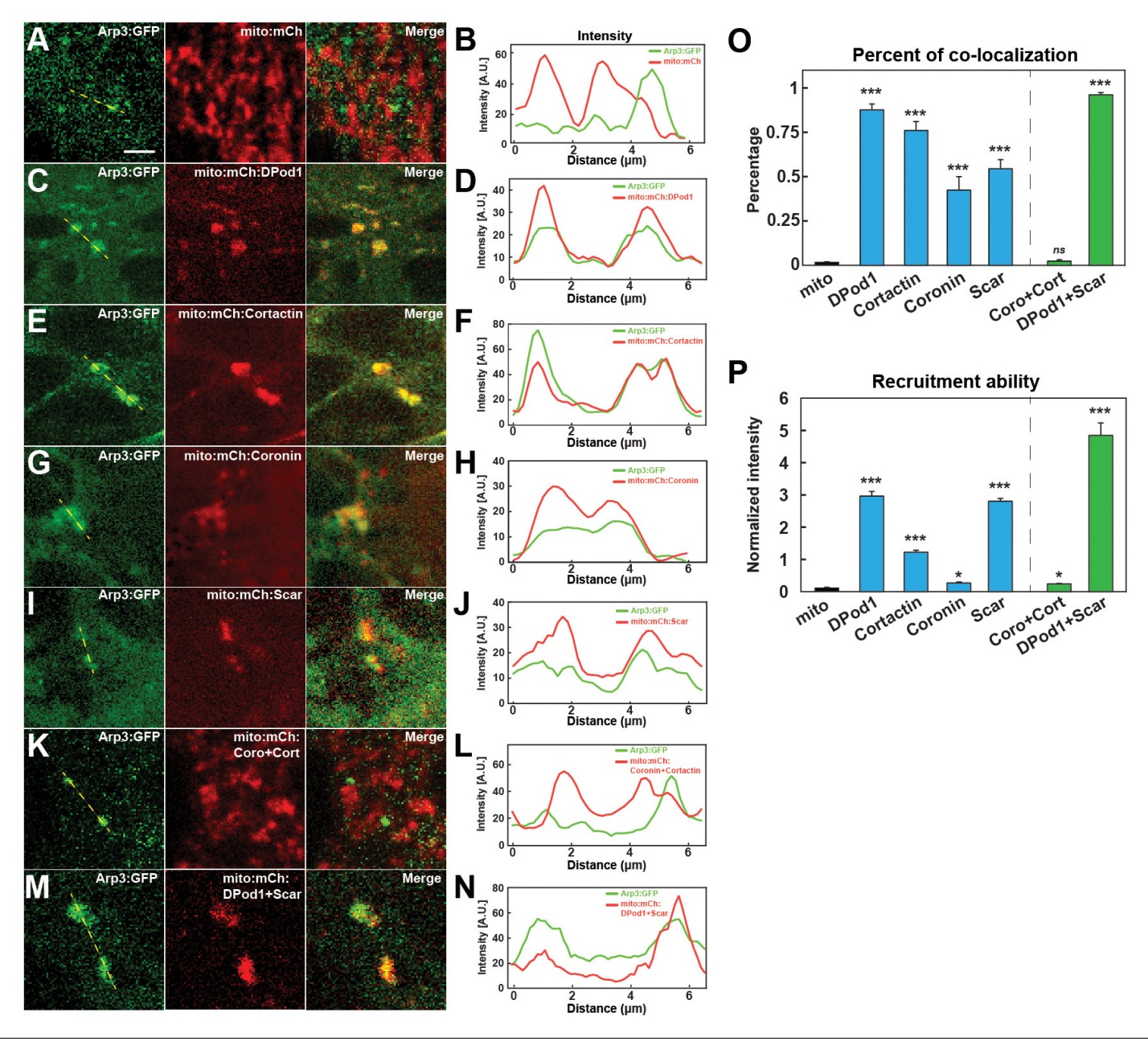

**Figure 6.** In vivo nucleator recruitment strengths of actin regulatory proteins. (A, C, E, G, I, K, M) Images of CRISPR Arp3:GFP with mito-tagged mCherry control (A) or mCherry:ANRPs (C, E, G, I, K, M) in cells at stage 12. Scale bar = 3 μm. (B, D, F, H, J, L, N) Intensity line plots of CRISPR Arp3: GFP and different mito-ANRP measured from yellow dashed lines in (A, C, E, G, I, K, M). (O) Percent of Arp3:GFP-positive compartments that colocalize with mito-ANRP puncta. Mito-control tag (no ANRP): n = 150; N = 4; DPod1: n = 338, N = 6; Cortactin: n = 198, N = 3; Coronin: n = 80, N = 4; Scar: n = 74, N = 6; Coro + Cort: n = 28, N = 4; DPod1 + Scar: n = 282, N = 4. Similar data trends were observed after calculation of Pearson correlation coefficients with Mito-control tag (no ANRP): r = −0.036 ± 0.025; DPod1: r = 0.625 ± 0.033; Cortactin: r = 0.492 ± 0.039; Coronin: r = 0.234 ± 0.028; Scar: r = 0.313 ± 0.024; Coro + Cort: r = 0.0231 ± 0.012; DPod1 + Scar: r = 0.551 ± 0.020. Reported Pearson values are (mean) ± (standard error of mean). (P) Arp3 recruitment ability (normalized GFP:mCherry intensity ratio in Arp3:GFP-positive mito-ANRP compartments) by mito-tagged DPod1, Cortactin, Coronin, and Scar. DPod1: n = 26, N = 3; Cortactin: n = 29, N = 3; Coronin: n = 27, N = 4; Scar: n = 40, N = 3; Coro + Cort: n = 25, N = 4; DPod1 +Scar: n = 46, N = 5.

The online version of this article includes the following source data and figure supplement(s) for figure 6:

**Source data 1.** Mito-localization source data.

**Figure supplement 1.** Mito-tagged ANRPs can direct F-actin polymerization.

branching points (*Cai et al., 2008*). These results are consistent with a combinatorial model for structuring the apical cortex in which DPod1 supports overall amounts of actin filaments, Coronin supports very early actin cap growth, and Cortactin and Scar promote mid-to-late cap growth and maintenance, although there are varying degrees of overlap in these functions. We also observed that several ANRPs (Carmil, Wasp, and Wash) appeared to have limited or no function at regulating actin cap formation at these stages, as judged both at the level of localization and function, although it is possible that residual gene function after shRNA disruption could obscure potential phenotypes.

We also comprehensively quantified apical actin dynamics when the major Formin and Arp2/3 actin networks are disrupted. Earlier works in the fly embryo suggested that the actin cap is largely dependent on Arp2/3 function, while the filamentous actin supporting ingressing furrows is largely Diaphanous/Formin driven (*Grosshans et al., 2005*; *Cao et al., 2010*; *Zhang et al., 2018*). Our results are broadly consistent with this viewpoint, although they also point to a lesser, but still substantial, Formin function in the cap. Recent work has suggested that Formin proteins and Arp2/3 complex function possess an intriguing interplay in apical caps, with Diaphanous-based actin bundles being displaced by Arp2/3 actin nucleation function (*Jiang and Harris, 2019*), and other data has shown a degree of interdependence between Arp2/3 and Formin network function (*Suarez et al., 2015*; *Chan et al., 2019*), although our measurements suggest that the *levels* of Diaphanous and Arp2/3 are not greatly changed after disruption of the counterpart network. Given the depth of the actin cap disruption in Arp2/3 compromised embryos, we cannot rule out that the developmental regulation of the phases of actin behaviors is altered in this background, though we would favor a more direct mechanical role of Arp2/3 in directing F-actin formation.

Our results also reveal a potential deep competition between the Diaphanous/Formin and Arp2/3 networks over the available G-actin pools. Somewhat surprisingly, actin fluorescent recovery times were approximately twice as fast when either network was compromised. Although either increases in filament turnover or filament assembly could explain these faster recovery times, the fact that chronic disruption of Diaphanous and Arp2/3 function (both of which are implicated in directing nucleation and filament growth) led to faster network recoveries is suggestive, to us, that this illustrates the degree to which G-actin availability limits filament assembly. These results would also be consistent with studies in *S. pombe* that have similarly suggested an upregulation of specific Formin or Apr2/3 networks when one network is disrupted (*Burke et al., 2014*). However, it should be noted that Coronin and DPod1 have each been implicated in regulating actin turnover, in addition to potential roles in regulating growth and nucleation (more on this below; *Cai et al., 2008*; *Gandhi and Goode, 2008*; *Mikati et al., 2015*). Contrary to this possibility, disruption of actin destabilizing Cofilin did not produce faster FRAP recovery. We find these results intriguing, as even given the volume of the *Drosophila* embryo ($9.02 \times 10^6$ µm$^3$; *Markow et al., 2009*) and the relatively few actin caps (~500–2000 caps during cycles 10–12) present in the early cortical cycles (i.e., per unit volume), these data suggest that local concentrations of G-actin can still become limiting at the cortex. Finally, we would note that the observed actin FRAP recovery times are faster than has been previously observed (*Cao et al., 2010*), although this may be due to differences in the measured syncytial cycle (cycle 11 versus cycle 12) or other methodological/instrumental differences.

To test the relative Arp2/3 recruiting potencies of the ANRPs, we chose to employ an ectopic relocalization strategy (*Wong and Munro, 2014*). This mito-tag technique has the advantage of testing factors in an intact tissue and cytoplasm, as opposed to artificially, buffered conditions in vitro. Although this approach does not differentiate between direct and indirect interactions, it interestingly revealed that DPod1 most potently recruited Arp2/3 to ectopic sites at the mitochondria, which correlates well with the importance of DPod1 for F-actin intensities at the cortical actin cap. Coronin had the weakest recruiting ability, possibly suggesting a primary role for Coronin in the regulation of Cortactin function and consistent with studies that suggest a complicated, and at times contradictory, function in Arp2/3 regulation (reviewed in *Gandhi and Goode, 2008*, and discussed below). Interestingly, when Scar was found to colocalize with Arp3:GFP, it was a very potent recruiter of Arp3/F-actin, but only a subset of mitochondrial Scar appeared active (~50% colocalization with Arp3:GFP). Embryos with disrupted DPod1, Cortactin, and Scar function also showed changes in actin stability and recovery rates (as indicated by immobile fractions and T50s) that mimicked the changes observed when Arp2/3 complex function was compromised. The partial colocalization of mito-tag Scar with Arp3:GFP additionally suggests a possible regulation and/or partial activation of Scar, which may be limiting in terms of Scar function and may explain why DPod1 is the

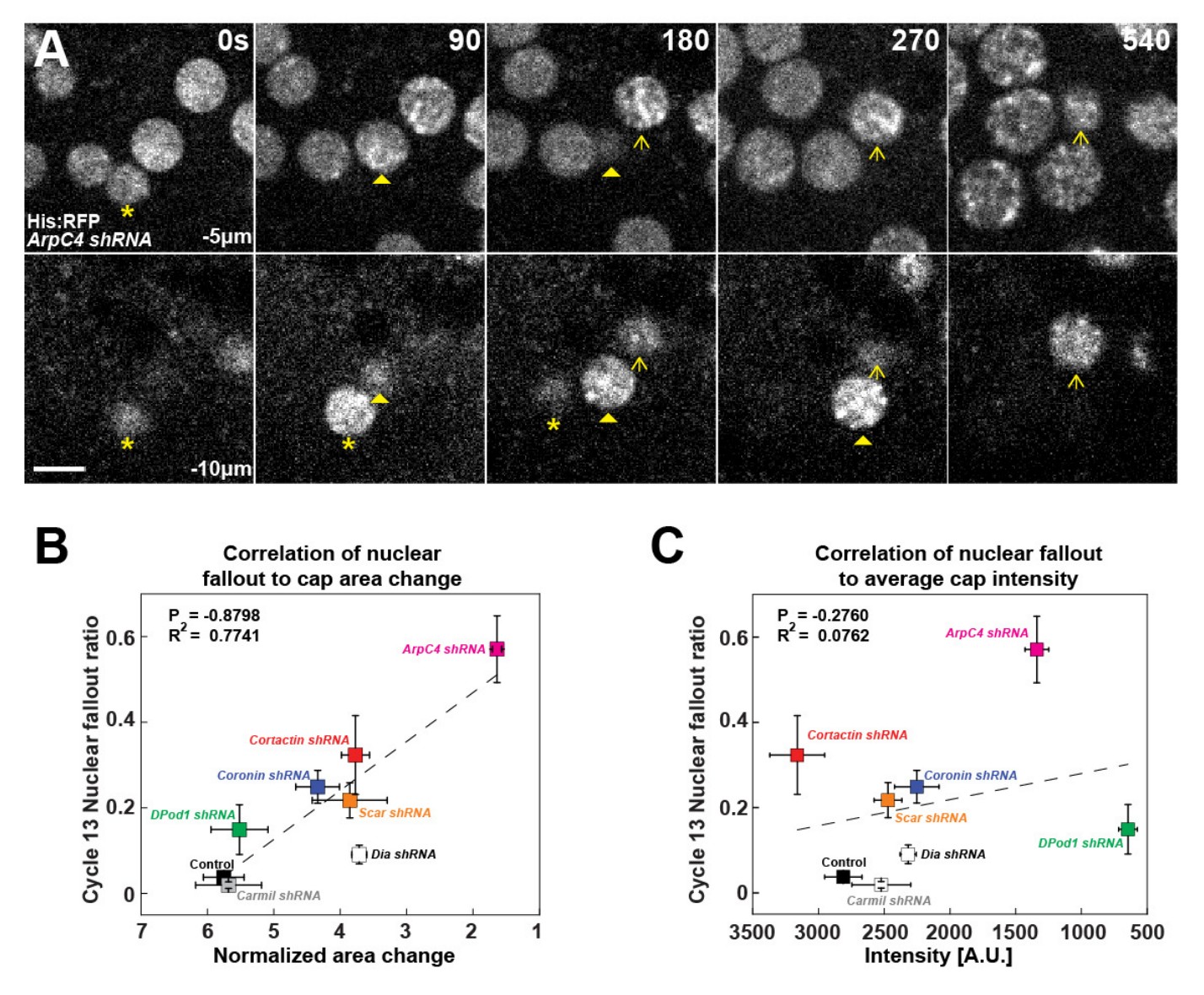

**Figure 7.** Requirement for filamentous actin cap ANRP function in anchoring embryonic nuclei. (**A**) Nuclei (marked by Histone:RFP) lose apical anchorage and fall into the embryonic interior in *ArpC4 shRNA* embryos during cycle 12 at t = 0 s, 90 s, 180 s, 270 s, and 540 s. Medial z-layer (−5 μm from apical most portion of embryo) indicates plane of normal nuclear positioning, and basal layer (−10 μm) images are shown. Asterisk, arrowhead, and arrow indicate individual falling-out nuclei. Scale bar = 5 μm. (**B**) Correlation of nuclear fallout rates to cap area expansion rates (t = 0–120 s) in indicated backgrounds (cycle 13 embryos). (**C**) Correlation of nuclear fallout rates to average actin cap intensities (t = 120 s) in indicated backgrounds (cycle 13 embryos). (**B, C**) Control (n > 12, N > 3), *Dia shRNA* (n = 10, N = 3), *ArpC4 shRNA* (n = 11, N = 3), *DPod1 shRNA* (n = 11, N = 3), *Coronin shRNA* (n = 9, N = 3), *Cortactin shRNA* (n = 12, N = 3), *Scar shRNA* (n = 11, N = 3), and *Carmil shRNA* (n = 9, N = 3). Dashed lines indicate linear regression fitting. P: Pearson's correlation coefficient, $R^2$: coefficient of determination.

The online version of this article includes the following source data for figure 7:

**Source data 1.** Nuclear fallout source data.

most potent regulator of actin network function at these stages despite the similar strength with which Scar appears capable of recruiting Arp2/3 complex function. It may also be that this regulation is limiting in the relocalization assay and suggests one of the caveats to this approach (namely, that although this technique had the advantage of being in vivo, it still represents recruitment to an unnatural compartment that may have its own limitations imposed by the presence or absence of upstream signals and lipid bilayer composition). Nevertheless, this approach is a nicely

**Table 2.** Stocks and genetics.

| Stocks | Source | Identifier |
|---|---|---|
| General stocks | | |
| P[mat-tub-Gal4] mat67 | D. St Johnston | |
| P[mat-tub-Gal4] mat15 | D. St Johnston | |
| UAS:mCherry:MoesinABD | T. Millard | |
| Histone:RFP | BDSC | BDSC 23650 III; BDSC 23651 II |
| UAS:GFP:Act88F | BDSC | BDSC #9253 |
| Wasp:sGFP | VDRC | VDRC #318474 |
| Wash:GFP | BDSC | BDSC #81644 |
| YFP:mito | BDSC | BDSC #7194 |
| UAS:mCh:mitoOMM | BDSC | BDSC #66532, 66533 |
| nos-Cas9 | Bestgene | NIG-FLY #CAS-0001, CAS-0003 |
| w1118 | Bestgene | |
| shRNA (Valium) lines | | |
| ArpC4 shRNA | DRSC/TRiP | BDSC #41888 |
| Dia shRNA | DRSC/TRiP | BDSC #35479 |
| DPod1 shRNA | DRSC/TRiP | BDSC #41705 |
| Coronin shRNA | DRSC/TRiP | BDSC #40841 |
| Cortactin shRNA | DRSC/TRiP | BDSC #44425 |
| Carmil shRNA | DRSC/TRiP | BDSC #41686 |
| Scar shRNA | DRSC/TRiP | BDSC #51803 |
| Wasp shRNA | DRSC/TRiP | BDSC #51802 |
| Wash shRNA | DRSC/TRiP | BDSC #62866 |
| Cofilin shRNA | DRSC/TRiP | BDSC #33670 |
| DPod1 shRNA 2 | VDRC/KK | VDRC #108886 |
| Coronin shRNA 2 | VDRC/KK | VDRC #109644 |
| Cortactin shRNA 2 | VDRC/KK | VDRC #105289 |

complementary technique to in vitro biochemical measurements and may provide a useful alternative approach for assaying protein recruitment abilities in the early fly embryo in vivo.

Finally, although much of our focus has been on the ANRPs in terms of guiding F-actin nucleation through the Arp2/3 complex, it should be pointed out that these regulators have been implicated in other actin-related processes (such as the control of filament branching and turnover) that may be responsible for their relative effects on actin growth and intensities. Our functional results illustrate the final outcomes of disrupting ANRP function on actin morphologies, and, combined with the mito-tag assay, suggest the strength of Arp2/3-dependent regulation, but it is clear that several of these proteins have been implicated in additional biochemical processes other than Arp2/3 activation. For example, Coronin has been observed to both promote and inhibit Arp2/3 function, as well as directing F-actin turnover through cofilin/GMF function (*Gandhi and Goode, 2008*; *Mikati et al., 2015*). The Scar/Wasp/Wash family of proteins is typically viewed as directly activating Arp2/3 nucleating activities (reviewed in *Molinie and Gautreau, 2018*), and Cortactin can also activate nucleation at high concentrations, but additionally inhibits Arp2/3 debranching after nucleation has begun (*Weaver et al., 2001*; *Uruno et al., 2001*; *Weaver et al., 2002*; *Cai et al., 2008*). In other systems, Cortactin and Coronin have been found to compete in either stabilizing or destabilizing Arp2/3 branch points, and Cortactin is often preferentially found in newer filaments of migrating lamellipodia (*Cai et al., 2008*). It is tempting to speculate that this new-branch stabilizing function of Cortactin could be a reason why Cortactin has been selected to support the edge outgrowth of the caps.

**Table 3.** Reagents.

| Reagent | Source | Identifier |
|---|---|---|
| Antibodies and dyes | | |
| Rabbit anti-GFP | Invitrogen | A11122 |
| Mouse anti-dsRed | Clontech | 632393 |
| Alexa Fluor Goat anti rabbit 488 | Invitrogen | A11034 |
| Alexa Fluor Goat anti mouse 568 | Invitrogen | A11031 |
| Alexa 568-Phalloidin | Invitrogen | Cat# A12380 |
| Alexa 647-Phalloidin | Invitrogen | Cat# A22287 |
| Rabbit anti-Dia | Wasserman lab | |
| Chemicals and kits | | |
| Halocarbon oil 27 | | Cat# H8773 |
| Halocarbon oil 700 | | Cat# H8898 |
| | Paraformaldehyde | Electron Microscopy Sciences |
| Cat# 15714 | | |
| ProLong Gold | Invitrogen | Cat# P36931 |
| Jasplakinolide | Santa Cruz Biotech | Cat# sc202191 |
| Latrunculin B | Sigma | Cat# L5288 |
| QIAShredder | QIAGEN | Cat# 79654 |
| Quick-RNA MicroPrep | Zymo Research | Cat# R1050 |
| QuantiTech Reverse Transcription Kit | QIAGEN | Cat# 205310 |
| QuantiTech SYBR Green RT-PCR | QIAGEN | Cat# 204141 |
| Q5 site-directed mutagenesis EZNA insect DNA kit | NEB Omega Biotek | Cat# E0554S Cat# D0926-01 |
| Software | | |
| iQ5 | Bio-Rad | bio-rad.com |
| FIJI/ImageJ | *Schindelin et al., 2012* | Fiji.sc |
| Micromanager 1.4 | *Edelstein et al., 2014* | micro-manager.org |
| OriginPro | OriginLab | originlab.com |
| Photoshop | Adobe | adobe.com |
| Illustrator | Adobe | adobe.com |
| Peptides and | oligonucleotides | |

*Table 3 continued on next page*

*Table 3 continued*

| Reagent | Source | Identifier |
|---|---|---|
| Coronin peptide for antibody | GenScript | CLPAKKAGNILNKPR |
| TOM70-HA | S. Munro lab | |
| qPCR primers | | |
| Sqh(MRLC) | QuantiTect | Cat# QT00499065 |
| Rh3 | QuantiTect | Cat# QT00978481 |
| DPod1 set1 | QuantiTect | Cat# QT00499464 |
| DPod1 set2 | Eurofins | 5'-TCCTCACCAAGAACCACTGC |
| | Eurofins | 5'-GTGGGTGGGAACAGATCGTC |
| Coronin set1 | QuantiTect | Cat# QT00940737 |
| Coronin set2 | Eurofins | 5'-ACAGGCTTCAACCGTAGCTC |
| | Eurofins | 5'-GAACATTACGCCGTTGGACG |
| Cortactin set1 | QuantiTect | Cat# QT00979020 |
| Cortactin set2 | Eurofins | 5'-TTCGGAGTGCAAGAGGATCG |
| | Eurofins | 5'-GCACTCCAAATTTGCCTCCG |
| Arp14D | QuantiTect | Cat# QT00923419 |
| ArpC1(sop2) | QuantiTect | Cat# QT00936222 |
| Dia set1 | QuantiTect | Cat# QT00939477 |
| Dia set2 | Eurofins | 5'-CAAATCGAAGGAGGAGCGACA |
| | Eurofins | 5'-CCCATTCTGCAGGTATTCCAC |
| Wasp set1 | QuantiTect | Cat# QT00984641 |
| Wasp set2 | Eurofins | 5'-ATGGCATGGAGGTGGTCAAG |
| | Eurofins | 5'-TTACGCGTCTCTATGGTGGC |
| Scar set1 | QuantiTect | Cat# QT00934584 |
| Scar set2 | Eurofins | 5'-ACGATCCATAGAACCCGTGC |
| | Eurofins | 5'-GGCGAATGATGTTCGTCAGC |
| Carmil set1 | Eurofins | 5'-CCACTGGTGGGTCGTAAGTC |
| | Eurofins | 5'-GGCATAGACGTCTCCTCAGC |
| Carmil set2 | Eurofins | 5'-GCTGAGGAGACGTCTATGCC |
| | Eurofins | 5'-ATAACACTACCCTCGCCTGC |
| Wash | Eurofins | 5'-GCGTAGGAAGAGTGTGGGAC |
| | Eurofins | 5'-GTGATGGAATTGCGCTCGTC |
| Guide RNAs for CRISPR | | |
| Arp3:GFP | | |
| chiRNA1 | Eurofins | 5'-CTTCGCTATCAGGTGTGTCACACGA |
| | Eurofins | 5'-AAACTCGTGTGACACACCTGATAGC |
| chiRNA2 | Eurofins | 5'-CTTCGCCAGTTCAACCCCCTATCTA |
| | Eurofins | 5'-AAACTAGATAGGGGGTTGAACTGGC |
| GFP:Cortactin | | |
| chiRNA1 | Eurofins | 5'-CTTCGGGGCCGACAAAGCCGGATC |
| | Eurofins | 5'-AAACGATCCGGCTTTGTCGGCCCC |
| chiRNA2 | Eurofins | 5'-CTTCGGTGGCCTGAATCTGGTGAC |
| | Eurofins | 5'-AAACGTCACCAGATTCAGGCCACC |

*Table 3 continued on next page*

*Table 3 continued*

| Reagent | Source | Identifier |
|---|---|---|
| GFP:DPod1 | | |
| chiRNA1 | Eurofins | 5'-CTTCGAGCGACTGAGAGGGAGCCAC |
| | Eurofins | 5'-AAACGTGGCTCCCTCTCAGTCGCTC |
| chiRNA2 | Eurofins | 5'-CTTCGCGATGTTGTTACCGTACGTC |
| | Eurofins | 5'-AAACGACGTACGGTAACAACATCGC |
| DPod1 mutated PAM sites in homologous constructs | This study | 5'-CCACCGGACTAGTGACACTCGAC<br>5'-GCAGCGCACAACTGACACTCGAC |
| | This study | 5'-GTGGGCAGCTACCAGACGTACGG<br>5'-GTGGGCAGTTATCAAACCTATGG |

Vertebrate homologs of DPod1 (Coronin7) have also been shown to bind and regulate Cdc42 and/ or Rac function, suggesting one potential mechanism of actin/Arp2/3 regulation (*Swaminathan et al., 2015*; *Bhattacharya et al., 2016*). However, regardless of the above various activities, our results show the final products of these factors on the apical, cortical actin networks that form and position nuclei in the early fly embryo. It will be interesting in future experiments to begin to further examine the biochemical partners that may help mediate the activities observed in this study.

# Materials and methods

## Fly stock and genetics

All stocks were maintained at 25°C. Genotypes used in this study are listed in *Tables 1* and *2*. To generate endogenous GFP reporter constructs, the CRISPR/Cas9 system was used to knock-in an N-terminal (downstream of ATG site) or C-terminal (upstream of stop codon) GFP tag through the use of a donor construct with 1 kb or 1.5 kb homologous sequences flanking GFP. The homology donors were constructed in pBluescript SK(-). The upstream and downstream guide RNAs were designed in flyCRISPR (https://flycrispr.org/) and inserted into pU6-BbsI-chiRNA. Genomic PAM sites were pre-verified by DNA sequencing to avoid single-nucleotide polymorphisms present in different *Drosophila* lines. Donor constructs (500 ng/μL) and guide RNA constructs (100 ng/μL) were mixed and injected into nos-Cas9 expressing embryos (BestGene). Potential insertions were balanced, and flies were screened by genomic PCR (Platinum Taq DNA polymerase, Invitrogen) after genome extraction from larva or adults (E.Z.N.A insect DNA kit, Omega Biotek).

To generate UAS GFP-tagged fly stocks, N-terminal or C-terminal eGFP was inserted into pUASp or pUASt along with the coding sequence for a given gene. Mito-tagged constructs were made by inserting Tom70-HA (generous gift of S. Munro lab) at the N-terminus of mCh:ANRPs in pUASp. The constructs were injected into embryos (BestGene) for transgene recovery and balanced. UAS constructs were crossed to P[mat-tub-Gal4] mat67; P[mat-tub-Gal4] mat15 (mat 67; 15) maternal drivers for Gal4-driven expression. To knockdown gene function, shRNA lines were also crossed to mat-tub-Gal4 lines, and females were recovered from either mat-67-Gal4; mat-15-Gal4 double Gal4 lines for high shRNA expression or to individual mat-67-Gal4 or mat-15-Gal4 for moderate shRNA expression.

## Microscopy and time-lapse imaging

Spinning-disk confocal microscopy was performed on a Zeiss/Solamere Technologies Group spinning-disk with a 63 × 1.4 NA objective lens (image stacks were acquired every 5 s and were composed of 15 z-layers with 0.3 μm z-steps), or Olympus Fluoview FV3000 confocal laser scanning microscope with 40× or 60× 1.35 NA objective lens (images acquired every 5 s at 12 ms/pixel exposure settings). Embryos were collected on yeasted apple juice agarose plates. After dechorionation

in 50% bleach, embryos were transferred to an air-permeable membrane and mounted in Halocarbon 27 oil (Sigma). A coverslip was placed on embryos for live-imaging. For FRAP experiments, Olympus Fluoview FV3000 confocal laser scanning microscope with 40× or 60× 1.35 NA objective lens was used. Images were acquired every 1 s at 2 ms/pixel exposure settings. For drug injection, after dechorionation embryos were glued on a coverslip and dehydrated for 12–15 min, covered in Halocarbon oil 700 (Sigma). Jasplakinolide (20 µg/mL) and Latrunculin B (200 nM) were injected into embryos, followed by regular imaging protocol for FRAP and live-imaging. MicroManager 1.4, FIJI/ImageJ, and Olympus Fluoview software were used for image collection and analysis. All movies were acquired at 25°C.

## Embryo fixation, antibodies, immunostaining, and imaging

Dechorionated embryos were fixed at the interface of heptane and either 18.5% paraformaldehyde (Electron Microscopy Sciences) (*Postner and Wieschaus, 1994*) for 30 min for actin cap staining or 4% paraformaldehyde for 70 min for mito-tagged embryo staining, in 0.1 M sodium phosphate buffer (pH 7.4). The embryos were manually devitellinized and stained with rabbit anti-GFP (1:1000, Invitrogen) and/or anti-dsRed (1:500, Invitrogen). Alexa 546 or 647-phalloidin (1:200, Invitrogen), or secondary antibodies conjugated with Alexa 488 or 568 (Invitrogen), were used at 1:500. Coronin peptide antibody was used at 1:100 dilution. Coronin peptide antibody was generated by GenScript as peptide-KLH conjugation in New Zealand rabbits (sequence in *Table 3*). The affinity-purified antibody was used at 1:100 dilution (~10 µg/mL). Embryos were mounted in ProLong Gold (Life Technologies). Olympus Fluoview FV3000 confocal laser scanning microscopy was used for immunostained embryos imaging. Exposure settings of 8 or 12 ms/pixel were used for image acquisition.

## Actin cap dynamics measurements

Apical cap dynamics were measured by live-imaging embryos with the F-actin marker UAS:mCh:MoeABD. UAS:mCh:MoeABD displays very similar dynamics to directly labeled actin (*Figure 1—figure supplement 1*), suggesting that on/off rates of the moesin actin binding domain are much slower than actin assembly/disassembly rates, and mCh:MoeABD marks similar structures as phalloidin reveals (*Figure 1—figure supplement 1*; *Blankenship et al., 2006*). UAS:mCh:MoeABD has also been used extensively in *Drosophila* as an F-actin reporter (*Kiehart et al., 2000*; *Blankenship et al., 2006*; *Spracklen et al., 2014*). The measured apical cap region was determined by the region 0.9 µm (3 z-planes) below the apical most layer in which the embryo could be detected. The apical cap area was selected through manual segmentation based on the cap F-actin boundary after background subtraction. Area, average intensity (average intensity values of the measured pixels, and not summed intensities across the total cap), and standard deviation were quantified in FIJI/ImageJ. Normalization was used to reduce variability due to different initial absolute cap areas that occurs naturally in different embryos. To do so, cap measurement traces were aligned to the first time point of cap elongation phase when the centrosomes separate. Cap behaviors were then tracked backwards to the first moment that a new cap was clearly present (this is approximately t = −180 s from the alignment point at cycle 11), and cap area at this time was used as the initial t = 0 area. Cap areas were normalized to the cap size at cap initiation. This gave the most robust and reproducible measurements of changes in cap area.

## Intensity measurements

For Arp3:GFP intensity measurements, a circular region (7705 px$^2$, ~207 µm$^2$) was quantified in each cap with FIJI/ImageJ. For line intensity measurements, the intensity profiles were quantified in FIJI/ImageJ and smoothed by averaging three neighboring points.

## FRAP intensity measurements

Photobleaching was performed on an Olympus Flouview FV3000 confocal laser scanning microscope with 60 × 1.35 NA objective lens as described above. Five hundred and sixty-one nanometer laser photobleaching of ROIs were drawn to encompass single individual caps, or specific regions of caps (center versus periphery), and ROI data was used for analysis of actin recovery rates. Each single cap FRAP was collected from a single embryo (multiple cap FRAP data were not acquired from a given embryo). Images were acquired every 1 s at 2 ms/pixel exposure settings for a total of 60–120 s.

ImageJ was used for measuring cap total intensity from FRAP data. To minimize potential effects of focal plane or biological drift, two non-FRAP caps were measured and used for normalization. Normalized intensity = ((FRAPed cap intensity − background) * FRAPed cap area) / ((non-FRAPed cap intensity − background) * non-FRAPed cap area). To calculate the t50 and immobile fraction, we fitted the FRAP data to two-phase association in Graphpad Prism eight with $SpanFast = (Plateau-Y_0) (\%Fast*0.01)$; $SpanSlow = (Plateau-Y_0) (100-\%Fast*0.01)$; $Y=Y_0 + (SpanFast) (1-exp(-KFast*X)) + (SpanSlow) (1-exp(-KSlow*X))$. X is time in seconds; Y is total cap intensity over time. $Y_0$ is the cap total intensity immediately after FRAP (t = 0 s). We assumed that the fast half-time measurement indicates the fluorescence recovery by the surrounding G-actin pool addition and turnover, while the slow half-time was produced through the minor cap expansion and intensity increases during this short time period. Fast half-time measurements and fractions are reported as the measured t50 and immobile fractions. Immobile fractions = $(Y_{-1} - PlateauFast) / (Y_{-1} - Y_0)$. $PlateauFast = 2 * SpanFast * (1-exp(-KFast*t50)) + Y_0$. $Y_{-1}$ is the cap intensity 1 s before FRAP.

### Real-time PCR

shRNA lines were crossed to P[mat-tub-Gal4] mat67; P[mat-tub-Gal4] mat15. The F1 embryos were collected by standard protocol, shredded (QIAShredder, QIAGEN), and RNA extracted (Quick-RNA MicroPrep, Zymo Research). The RNA extracts were reverse transcribed (QuantiTech Reverse Transcription Kit, Invitrogen) and used for real-time PCR (QuantiTech SYBR Green RT-PCR, Invitrogen; Bio-Rad iQ5). The primers used for RT-PCR are listed in *Table 3*.

### Statistics and repeatability

A Shapiro-Wilk test was performed in OriginPro to test for statistical normality of data. Cap area, area change, expansion rate, intensity, heterogeneity, and heterogeneity change data were tested for statistical significance using Student's t-test for all normal data. ns: $p>0.05$; *: $p<0.05$; **: $p<0.005$; ***: $p<0.0005$. Each cycle 11 cap was measured for ~110 time points (every 5 s), with all measurements being quantified from at least nine individual caps from a minimum of three embryos. n represents the total number of individual structures measured, and N represents the total embryos tested. For mito-tag colocalization, mitochondrial puncta greater than 8 $\mu m^2$ were measured and puncta that shared a minimum of at least 10 pixels in common were determined as possessing colocalization. Pearson coefficient calculations were performed with ImageJ Coloc two plug-in using Costes threshold regression with PSF = 3.0 and Costes randomization = 10. Data for the 0–60 s periods of rapid expansion fits the following exponential function, $Y = 1.642 * exp*(0.09733*X)$, with an R value of 0.9139 (as calculated in Graphpad Prism 8).

### Image editing and figure preparation

Spinning-disk and laser scanning confocal microscopy images were edited by FIJI/ImageJ and Adobe Photoshop. Images were uniformly leveled for optimal channel appearance except where noted. Actin cap curves (average values and errors), bar graphs (average values and errors), box and whisker plots (boxes as 25–75% values, whiskers as minimal and maximal values, and lines in the boxes as median), and other graphs were made in OriginPro. Error bars are shown as S.E.M. Figures were prepared and labeled in Adobe Illustrator.

## Acknowledgements

The authors declare no competing financial interests. We thank members of the Blankenship lab for critical reading and constructive comments on the manuscript. This work was supported by grants from the NIH NIGMS: R01GM127447 and R15 GM126422 to JTB.

## Additional information

### Funding

| Funder | Grant reference number | Author |
| --- | --- | --- |
| National Institute of General Medical Sciences | R01GM127447 | J Todd Blankenship |

| National Institute of General Medical Sciences | R01GM126422 | J Todd Blankenship |

The funders had no role in study design, data collection and interpretation, or the decision to submit the work for publication.

### Author contributions
Yi Xie, Conceptualization, Formal analysis, Investigation, Visualization, Methodology, Writing - original draft, Writing - review and editing; Rashmi Budhathoki, Formal analysis, Investigation, Methodology; J Todd Blankenship, Conceptualization, Formal analysis, Supervision, Funding acquisition, Investigation, Methodology, Writing - original draft, Writing - review and editing

### Author ORCIDs
J Todd Blankenship (iD) https://orcid.org/0000-0001-8687-9527

### Decision letter and Author response
Decision letter https://doi.org/10.7554/eLife.63046.sa1
Author response https://doi.org/10.7554/eLife.63046.sa2

## Additional files
### Supplementary files
• Transparent reporting form

### Data availability
All data generated or analyzed during this study are included in the manuscript and supporting files. Transgenic stocks have been made freely available.

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
