## [Decision Letter]

**Acceptance summary:**

This work shows the molecular complexity involved in the assembly of actin networks, and this in-depth characterization will be important for the field. The FRAP data showing faster recovery of actin networks after regulator disruption is interesting. It questions how this excessive monomeric actin reservoir is maintained and consumed under these conditions, and what it represents in terms of size control for these actin networks. Future modeling would be greatly beneficial to understand how actin fluxes are controlled in this system.

**Decision letter after peer review:**

Thank you for submitting your article "Combinatorial deployment of F-actin regulators to build complex 3D actin structures in vivo" for consideration by *eLife*. Your article has been reviewed by 3 peer reviewers, including Alphee Michelot as the Reviewing Editor, and the evaluation has been overseen by Anna Akhmanova as the Senior Editor. The following individual involved in review of your submission has agreed to reveal their identity: Anne Cecile Reymann (Reviewer #3).

The reviewers have discussed the reviews with one another and the Reviewing Editor has drafted this decision to help you prepare a revised submission.

Summary:

Xie and Blankenship present here a characterization of the apical actin architectures during the syncytial cell cycles of the fly early embryo. These actin architectures undergo a series of rapid and transient reorganization correlated with cell cycle. The authors examine the functions of one formin (diaphanous), of the Arp2/3 complex, and of 7 other proteins (called here ANRPs) involved in the activation of the Arp2/3 complex and/or in various other actin regulation activities. Their results reveal a clear complexity, as these proteins display different effects on actin caps dynamics and in anchoring embryonic nuclei.

This work also includes information of ANRPs localization at actin caps, and their relative role in recruiting the Arp2/3 complex. Among those observations, the authors find a clear antagonism between cortactin and coronin, which was shown in previous studies but is nicely confirmed here.

In the last part of this work, which is more debatable, the authors performed FRAP experiments on actin caps, and find surprisingly that recovery of actin signals are faster in mutant conditions than in WT. These results, together with the fact that treatment with LatB or Jasp increases on the contrary characteristic times of recovery, suggest to the authors that the pool of monomeric actin might be increased in mutant conditions, potentially revealing a competition between actin networks for a limited pool of monomeric actin.

Overall, the Reviewers agree that tools and results presented here will be valuable for the community. The experimental procedures are adapted and well performed, and revisions should require limited experimental work. However, the Reviewers regret some lack of explanation on the quantification and statistical analysis of the experiments, notably regarding the lack of detailed explanation on the procedures (segmentation procedure, measure of intensity, calculation of errors, FRAP analysis, colocalization etc).

Essential revisions:

1. Quantification of the data:

Could the authors use a more automatized or robust method, such as via an autocorrelation function, to measure the size of the caps without the bias of using a threshold or a manual segmentation? Could they also provide videos for each conditions as Supplementary data?

How initial time points of cap formation are determined is not clear. Segmentation or thresholding methods to obtain area and intensities are not clearly described. Measured Area is often shown as normalized over the initial area, but this initial time point seems prone to fluctuations or errors of segmentation.

Regarding actin intensity: is it a total intensity or a mean density which is important as the area is indeed changing? Important to know if we are considering total actin filaments change or local density changes. Making this distinction would also be interesting while discussing shRNA results. There is also no mention on the impact of bleaching even though 5s interval of 4D acquisitions are performed over several minutes.

From this quantification and the definition of clear parameters, it should also appear clearly why the authors report that cap dynamics share similar features in cycle 10-13. From an observer's eye, it seems on the contrary that some of these phases are absent from certain cycles. Also, Figure 2C, 3B,E,G,I,K and M are not clear, because each phases are defined based on WT conditions only. It would be important to report how phase durations are modified in shRNAs lines.

Finally, some results from this quantification are curious. For example, between time 0 and 60s in WT, cap intensity only increases moderately (Figure 1G), while cap area increases a lot (Figure 1D). This suggests that actin density should decrease a lot, which seems opposite to what is observed in Figure 1A.

2. Statistical analysis:

In Figures 5 and S7 the number of caps analyzed seems fairly small (n=3 or 4 in some cases) for FRAP experiments. The authors should increase their sample size and show the degree of variability in the FRAP parameters that they measured using box plots and standard deviations (rather than standard error).

The authors should also clarify what the variation observed for cells within a single embryo is vs. the variation for all embryos considered in this study.

3. The authors should report the efficiency of silencing in the shRNA experiments. This would enable readers to evaluate if residual activity of these proteins is expected or not. When some shRNA as Carmil, Wasp or Wash have little impact on cap dynamics, could it be just a question of efficiency of silencing? In the absence of such measurements, we would be more careful with the conclusions of these experiments. In some other systems too, wasp or wave silencing has little effect on cortical dynamics, to compensate gex-3 can be depleted and is more efficient. Did the author consider this option?

4. Based on the hollowed appearance of actin caps in Arp2/3 shRNA embryos, the authors propose that Arp2/3 may promote central actin assembly that then flows outwards. This seems easy enough to test, for example by photobleaching a region in the center of the cap and examining whether that photobleached region propagates outwards. The authors should conduct the same experiment upon Cortactin shRNA or Scar shRNA treatment, which seem to impede actin flow outwards (Figure 3H-I).

5. Regarding FRAP, the authors decided to use MOE:GFP as a proxy for actin dynamics. This rational is not well explained (line 266) and should be more justified.

For example, line 114: "This method of labeling has been used extensively in the *Drosophila* embryo, and well-represents endogenous filamentous actin dynamics while avoiding problems that occur when fluorescent proteins are directly attached to actin or other labeling paradigms". One could always argue that any fluorescent peptide binding to actin potentially impacts actin dynamics in some ways. What would be important for this study is to justify why this specific marker does not interfere for the specific observations made here. Additionally, it would be important to note that FRAP recovery times are equivalent to GFP:Act88F, indicating that recovery times represent rates of actin turnover rather than dynamics of MoeABD binding/unbinding.

Note that in figure S2 one could show the recovery curves in addition to the extracted value to justify the similarity of the process. Comparison with other published data is not mentioned. For example, injected rhodamine actin followed during cycle 12 (2010 Cao et al. Current Biology) values are t_1/2_ = 18.9{plus minus}1.7 s and 87.3% of recovery (so 13% immobile). So more than twice the t_1/2_ value presented here (8.2s). The quantification is also not well documented. What intensity is used? Is it normalized to pre-bleach as well as to the value post bleach? Is there any curve fitting to exponential functions to extract the parameters? The authors could try to use some simple models of actin polymerization to estimate rates of assembly and disassembly from the FRAP data and try to get at which one is changing when Dia or Arp2/3 are lost (e.g. Kobb et al., MBoC, 2019). The authors should also discuss the choice of FRAP timing (performed at max Area 120s) so at a critical point in terms of assembly and reorganization of caps architectures.

Also, FRAP recovery images are provided only for WT condition, but not for shRNA embryos. The authors should provide these images in the Supplementary data.

6. There is an over simplification in this work, in considering that formin (diaphanous) and Arp2/3 networks assemble fully independently. There is clear evidence now, for example in lamellipodia, that formins and Arp2/3 can be synergistic, and it is not demonstrated that dia and Arp2/3 networks assemble independently here. The authors should take into account this possibility when discussing their results, and modify Figure 5G.

---

## [Author Response]

Essential revisions:1. Quantification of the data:Could the authors use a more automatized or robust method, such as via an autocorrelation function, to measure the size of the caps without the bias of using a threshold or a manual segmentation? Could they also provide videos for each conditions as Supplementary data?

We have been very interested in an automated approach, and have tried a variety of segmentation algorithms/approaches. However, largely due to the very spiky periphery of caps and the low SNR (signal-to-noise) at the cap edge under the current imaging conditions, we have not found a segmentation approach that determined cap outlines as well as our manual measurements. For now, this appears to be the most accurate way to capture these area. We would note that this is part of the reason for the moderate n numbers – careful manual measurements of cap sizes means that each cycle 11 cap takes ~2 hours to determine (for example, Figure 1 represents ~35 cumulative hours of manual measurements).

How initial time points of cap formation are determined is not clear. Segmentation or thresholding methods to obtain area and intensities are not clearly described. Measured Area is often shown as normalized over the initial area, but this initial time point seems prone to fluctuations or errors of segmentation.

This is a good point, and we have clarified this in the Methods. The reviewer(s) are correct that, especially in the early time points, individual cap behaviors have some degree of variability. To address this, we temporally aligned caps to the first moment of cap duplication when the centrosomes separate (beginning of the elongation phase). We then went back to the first moment that a new cap is present that will clearly become the cycle 11 cap (this is approximately t = -180 sec from the alignment point). Data on cap area was then the normalized to the size at this starting point to account for variation that is induced from caps either having smaller or larger initial starting sizes. This produced the best comparable data as shown by the error envelopes. Both absolute and relative cap areas are measured and are shown in Figure 1. Both absolute and relative areas show similar dynamics and are not significantly different (0 minute in Figure 3 E’, G’, I’, K’, M’). For example, cap areas dynamics of Cortactin shRNA in Figure 3I (relative area) and I’ (absolute area) are similar, but normalized area dynamics better showed how cap areas evolved over time with less variation due to absolute changes in initial starting cap sizes. Please note that we did not observe that any of our shRNA lines affected the timing of centrosome separation.

Regarding actin intensity: is it a total intensity or a mean density which is important as the area is indeed changing? Important to know if we are considering total actin filaments change or local density changes. Making this distinction would also be interesting while discussing shRNA results. There is also no mention on the impact of bleaching even though 5s interval of 4D acquisitions are performed over several minutes.

Thank you for pointing this out. We are reporting average densities and not summed total actin intensities – we clarify this in the Methods (Lines 567-568). In terms of the potential photobleaching through live-imaging, this is one of the advantages of the fly embryo – we rarely observe much in the way of photobleaching. We attribute this in part to: 1) the use of moderate laser powers and exposure time (100ms) to reduce bleaching, and 2) the large volume of the embryo, most of which is not in the illuminated focal plane. We have analyzed long live-imaging videos that covered cycles 10 through 13 (~1 hour of acquisitions) and did not find significant reductions/trends of changes in total actin intensities with our imaging parameters.

From this quantification and the definition of clear parameters, it should also appear clearly why the authors report that cap dynamics share similar features in cycle 10-13. From an observer's eye, it seems on the contrary that some of these phases are absent from certain cycles. Also, Figure 2C, 3B,E,G,I,K and M are not clear, because each phases are defined based on WT conditions only. It would be important to report how phase durations are modified in shRNAs lines.

Yes, we believe that there is a similarity of phases in each of the cell cycles, but agree that there are moderate differences, especially in the short cycle 10. We have clarified the language in the manuscript to match this. Also, to make this similarity of phasing more apparent, we have added image panels to the Supplemental Figure S2 better illustrating the cap behaviors in cycles 10, 12, and 13. It appears that the Expansion and Fragmentation phases are clearly present in all cycles. In terms of the stabilization and elongation phases, cycle 10 is very short, so the stabilization phase is less clear, but the stabilization and elongation phases (as distinguished by the separation/elongation of the cap into an elongated, and eventually separated, cap) also appear to clearly occur in cycles 11-13. This is less obvious in the simple graph of cap area measurements, but changes in the cap dimensions marking elongation are apparent in the new panels in Figure Supplement1-2E-G. In terms of changes in phased behaviors in the shRNA backgrounds, it appears that each of the phases are generally present in each of the backgrounds (at least it does not appear that a phase has been deleted or that the timing of the cell cycle is deeply disrupted) except for in the background where Arp2/3 function is disrupted. In the Arp2/3 background, the cell cycle is ~60 seconds shorter and there is an absence of phased behaviors – we have added a short description of this to the Discussion section on Arp2/3 function.

Finally, some results from this quantification are curious. For example, between time 0 and 60s in WT, cap intensity only increases moderately (Figure 1G), while cap area increases a lot (Figure 1D). This suggests that actin density should decrease a lot, which seems opposite to what is observed in Figure 1A.

Yes, we believe this goes to what reviewers asked in 1.3, the cap intensity we showed here is average intensity, meaning that the actin density only moderately (~30%) increase from 0-120s (Figure 1G).

2. Statistical analysis:In Figures 5 and S7 the number of caps analyzed seems fairly small (n=3 or 4 in some cases) for FRAP experiments. The authors should increase their sample size and show the degree of variability in the FRAP parameters that they measured using box plots and standard deviations (rather than standard error).

Thank you for pointing it out. We have repeated our FRAP and increased our sample size. The control, ArpC4, and Dia shRNA data have been replaced with box and distribution plots.

The authors should also clarify what the variation observed for cells within a single embryo is vs. the variation for all embryos considered in this study.

We only FRAP one cap in a given embryo. We have clarified this in the Methods section (“FRAP intensity measurements” section).

3. The authors should report the efficiency of silencing in the shRNA experiments. This would enable readers to evaluate if residual activity of these proteins is expected or not. When some shRNA as Carmil, Wasp or Wash have little impact on cap dynamics, could it be just a question of efficiency of silencing? In the absence of such measurements, we would be more careful with the conclusions of these experiments. In some other systems too, wasp or wave silencing has little effect on cortical dynamics, to compensate gex-3 can be depleted and is more efficient. Did the author consider this option?

Another good point. Single embryo Westerns are difficult to obtain consistent results, and antibodies do not exist for several of the silenced proteins, so we have used at least two independent shRNAs to show the specificity of the observed phenotypes – this has generally been the standard for work in the early fly embryo. Very similar effects are observed between independent shRNA lines (Figure Supplement 4-1), suggesting that knockdown efficacy in different lines with different shRNA target sequences produces representative and reproducible defects (Figure Supplement 4-1). There is also a good correlation between the localization and functional data – for example, little-to-no localization was observed for Carmil, Wasp, and Wash in the early embryo, which also fits with the lack of phenotypes observed after disruption of these genes. Valium20 and Valium22 shRNA constructs have worked efficiently in the embryo in our lab, but it remains a possibility that residual activity could remain that obscures a phenotype. We have added a short discussion of this possibility to the Discussion section (Lines 402-405).

4. Based on the hollowed appearance of actin caps in Arp2/3 shRNA embryos, the authors propose that Arp2/3 may promote central actin assembly that then flows outwards. This seems easy enough to test, for example by photobleaching a region in the center of the cap and examining whether that photobleached region propagates outwards. The authors should conduct the same experiment upon Cortactin shRNA or Scar shRNA treatment, which seem to impede actin flow outwards (Figure 3H-I).

Yes, we thought this was a good idea and tried similar experiments prior to our first submission. We also tried this again after receiving the reviewer comments – we were disappointed by our current confocal microscope systems ability to specifically FRAP small subregions of actin caps – this could be either because of system limitations or because of optical imaging limitations with the vitelline membrane and intervening perivitelline space causing too much light scattering. It may be the latter as this inability to FRAP small regions was consistent across two different photobleaching systems (a new Olympus Fluoview FV3000 laser scanning confocal system and a spinning disc-mounted Andor Micropoint system). We did get preliminary data consistent with the our and the reviewers’ expectations, but thought the diffuseness of the photobleached regions was not up to our publication standards.

Below is the data of region specific FRAP – as discussed above, we have not been satisfied with the quality of the FRAP, so prefer not to include it in the main manuscript. Peripheral regions show a slower recovery than central regions, and compromising Cortactin function disrupts the regional differences of the resulting smaller actin caps (n>10 for all).

5. Regarding FRAP, the authors decided to use MOE:GFP as a proxy for actin dynamics. This rational is not well explained (line 266) and should be more justified.For example, line 114: "This method of labeling has been used extensively in the *Drosophila* embryo, and well-represents endogenous filamentous actin dynamics while avoiding problems that occur when fluorescent proteins are directly attached to actin or other labeling paradigms". One could always argue that any fluorescent peptide binding to actin potentially impacts actin dynamics in some ways. What would be important for this study is to justify why this specific marker does not interfere for the specific observations made here. Additionally, it would be important to note that FRAP recovery times are equivalent to GFP:Act88F, indicating that recovery times represent rates of actin turnover rather than dynamics of MoeABD binding/unbinding.

Yes, always a concern – on the one hand directly labeling actin subunits can lead to altered dynamics and disruption of filamentous networks, but indirectly labeling actin has its own concerns. We strengthened our justification in lines 283-287. MoeABD:GFP/MoeABD:mCherry or Utrophin:GFP have generally been the standard for visualizing actin dynamics in the early embryo. Both rely on the recognition of filamentous actin, but appear to have the most moderate effects on actin networks in terms of the available labeling methods. This is why we have compared our imaged dynamics with those of directly labeled actin as well as phalloidin stained actin caps (newly combined Figure Supplement 1-1).

Note that in figure S2 one could show the recovery curves in addition to the extracted value to justify the similarity of the process. Comparison with other published data is not mentioned. For example, injected rhodamine actin followed during cycle 12 (2010 Cao et al. Current Biology) values are t_1/2_ = 18.9{plus minus}1.7 s and 87.3% of recovery (so 13% immobile). So more than twice the t_1/2_ value presented here (8.2s). The quantification is also not well documented. What intensity is used? Is it normalized to pre-bleach as well as to the value post bleach? Is there any curve fitting to exponential functions to extract the parameters? The authors could try to use some simple models of actin polymerization to estimate rates of assembly and disassembly from the FRAP data and try to get at which one is changing when Dia or Arp2/3 are lost (e.g. Kobb et al., MBoC, 2019). The authors should also discuss the choice of FRAP timing (performed at max Area 120s) so at a critical point in terms of assembly and reorganization of caps architectures.Also, FRAP recovery images are provided only for WT condition, but not for shRNA embryos. The authors should provide these images in the Supplementary data.

Thank you for pointing out this out – it’s true that our Methods section needed to be better detailed, which have done in the revised manuscript. We also adapted a method of FRAP measurements similar to that outlined by Kobb et al. (as suggested) with correction to adjacent, nonbleached caps. This should better adjust for changes in z-levels and/or timing. However, this did not significantly change our reported results. The faster recovery times we observe than has been seen in Cao et al. is interesting, and could be because of the different cycle times (cycle 11 vs cycle 12), methodology, or instrumentation – we comment briefly on this in the Discussion (Lines 438-441). We also added the requested additional FRAP images (Figure Supplement 5-1E).

6. There is an over simplification in this work, in considering that formin (diaphanous) and Arp2/3 networks assemble fully independently. There is clear evidence now, for example in lamellipodia, that formins and Arp2/3 can be synergistic, and it is not demonstrated that dia and Arp2/3 networks assemble independently here. The authors should take into account this possibility when discussing their results, and modify Figure 5G.

Yes, an interesting point – however, how to interpret this is an interesting, and complex, discussion. On the one hand, we actually see a faster recovery of the intact actin nucleating network when the other network is disrupted, which could argue against synergistic effects. Contrary to this, it is also true that decreases in overall actin levels could be due to possible indirect decreases in the surviving networks activity. This was interesting, so to look at it more closely, we examined Diaphanous cap intensities after Arp2/3 disruption, and Arp2/3 cap levels after Diaphanous disruption (new Figure Supplement 2-1B-E). Interestingly, neither Arp2/3 levels after Diaphanous disruption, nor Diaphanous levels after Arp2/3 disruption were significantly decreased. We have revised the Results and Discussion sections to mention these findings.